# Gradients Through Logarithmic Lens: Reformulating Optimization Dynamics

## Abstract

Optimization in deep learning remains a fundamental challenge, and developing techniques that improve training efficiency and enhance model performance is essential. We present a method for producing effective optimization frameworks, introducing the activation function LogLU (*logarithmic linear unit's*) and the optimizer ZenGrad (*zen represents smooth, gradients*), along with its momentum-based variant, M-ZenGrad, all of which incorporate the logarithmic formulation. We conducted extensive evaluations on benchmark datasets spanning vision and language tasks, demonstrating that each component individually enhances performance while collectively showcasing the advantages of the logarithmic approach. Additionally, ablation studies analyze the contribution of each method and careful hyperparameter tuning ensures robust and optimal performance, indicate the effectiveness of our logarithmic optimization framework across diverse tasks and datasets.

## 1 Introduction

Gradient-based optimization is the foundation of modern deep learning. It provides the process by which neural networks adjust their parameters and learn useful patterns from data Ruder (2016); Goodfellow et al. (2016). The way gradients flow through a model is critical, since it affects how information is passed across layers, how stable the training remains, and how quickly a model can converge Liu et al. (2025). When gradients vanish or explode, models struggle to train effectively, highlighting the importance of designing methods that preserve smooth and stable gradient flow Bengio et al. (1994); Zucchet & Orvieto (2024). Over the years, continuous improvements in both optimization algorithms and activation functions have been driven by the need to make gradient propagation more reliable. As networks grow deeper and tasks more complex, handling gradients effectively has become not just a technical detail, but a key factor that decides the success of large-scale learning systems Goodfellow et al. (2016); Nocedal & Wright (2006).

Activation functions Sharma et al. (2020) and optimizers form the backbone of how neural networks learn from data. It introduces the necessary non-linearity that allows models to represent complex relationships, while optimizers govern how gradient information is translated into parameter updates. These components have evolved to improve both the speed and stability of training Dubey et al. (2022). Carefully designed activations ensure smoother gradient propagation, reducing common issues such as vanishing or exploding gradients, and adaptive optimizers Sun (2020) leverage momentum to guide models toward more efficient convergence. These advancements have enabled modern networks to scale to deeper architectures Christobel & Suji (2024) and larger datasets. Building on this foundation, our work explores how incorporating logarithmic structures can provide a new lens for understanding and improving gradient during training.

In this work, we examine gradient-based learning through a logarithmic lens and introduce LogLU, an activation function designed to preserve smooth gradient propagation and enhance stability, alongside ZenGrad and its momentum-augmented variant, M-ZenGrad, which adapt parameter updates using logarithmic scaling. Theoretical analyses for both the activation function and the optimizers are provided in their respective sections (See Section 2 and Section 3). Extensive empirical evaluations are reported in Section 4, while hyperparameter tuning and ablation studies are reported separately in Section 5 and Section 4.4. Together, these investigations demonstrate that the embed-

ding of logarithmic principles provides a unified framework for understanding gradient behavior and optimization.

## 2 LOGARITHMIC LINEAR UNIT'S (LOGLU)

Let $f(x) : \mathbb{R} \to \mathbb{R}$ be the activation function defined in Equation 1, which applies distinct transformations depending on the sign of the input. Specifically, for inputs $x > 0$, LogLU acts as the identity function, thereby preserving linearity and facilitating stable gradient propagation. Conversely, for inputs $x \leq 0$, LogLU applies a negative logarithmic transformation shifted by one and offset by a small constant $\varepsilon$, which non-linearly compresses the input domain. This design ensures smoothly bounded gradients in the negative domain, promoting both stability and effective learning in deep neural networks.

$$f(x) = \begin{cases} x, & \text{if } x > 0, \\ -\log_e(-x + 1) + \varepsilon, & \text{if } x \leq 0. \end{cases} \tag{1}$$

**Proposition 2.1** (Gradient Bounds of LogLU). *Let, $f(x) = \text{LogLU}(x)$. Then the derivative $f'(x)$ is strictly positive and uniformly bounded above by 1; that is,*

$$0 < f'(x) \leq 1 \quad \text{for all } x \in \mathbb{R}.$$

*Proof.* We compute the derivative in each region:

For $x > 0$, we have $f(x) = x$, so $f'(x) = 1$. For $x \leq 0$,

$$f'(x) = \frac{d}{dx}[-\log_e(-x + 1)] = \frac{1}{-x + 1} \in (0, 1],$$

since $-x + 1 \geq 1$. Thus, $0 < f'(x) \leq 1 \quad \forall x \in \mathbb{R}$. $\qquad\square$

**Remark.** Proposition 2.1 shows that $0 < f'(x) \leq 1$ for all $x$, so the LogLU activation never induces exploding gradients. Moreover, since $f'(x) = 1/(1 - x) \to 0$ only as $x \to -\infty$, the derivative remains strictly positive for all finite pre-activations (raw linear responses $z = \sum_{i=1}^{d} w_i x_i + b$ before the nonlinearity is applied). Consequently, if pre-activations are bounded below by some negative value of $x$, then $1/(1 - x) \leq f'(x) \leq 1$, and the LogLU activation does not cause vanishing gradients under realistic bounded-input conditions Goodfellow et al. (2016).

**Proposition 2.2** (Lipschitz Continuity of LogLU). *Let the activation function $f(x) : \mathbb{R} \to \mathbb{R}$ be defined as above. Then* LogLU *is Lipschitz continuous on $\mathbb{R}$ with Lipschitz constant*

$$L = \sup_{x \in \mathbb{R}} |f'(x)| = 1.$$

*Proof.* By Proposition 2.1, it holds that

$$0 < f'(x) \leq 1 \quad \text{for all } x \in \mathbb{R}.$$

Since LogLU is differentiable with uniformly bounded derivative, the Mean Value Theorem implies that for any $x, y \in \mathbb{R}$, there exists $c$ between $x$ and $y$ such that Bednarczuk & Rutkowski (2021)

$$|f(x) - f(y)| = |f'(c)| \cdot |x - y|.$$

Using the bound on the derivative, it follows that

$$|f(x) - f(y)| \leq |x - y|.$$

Hence, LogLU is Lipschitz continuous with Lipschitz constant Xu & Zhang (2024)

$$L = \sup_{x \in \mathbb{R}} |f'(x)| = 1.$$

These results highlight important theoretical properties of the LogLU activation function. The fact that the derivative is strictly positive and uniformly bounded ensures that the function is smooth across its entire domain. In addition, the Lipschitz continuity with constant $L = 1$ guarantees that LogLU responds to changes in input in a controlled and stable manner. These properties contribute to consistent gradient flow during optimization. $\qquad\square$

# 3 OPTIMIZER

## 3.1 VANILLA ZENGRAD

Let $\mathbf{w}_t \in \mathbb{R}^d$ denote the parameter vector at optimization step $t$, and let $\gamma > 0$ denote the base learning rate. The instantaneous gradient of the loss function $\mathcal{L}(\mathbf{w})$ at step $t$ is given by $\nabla_{\mathbf{w}}\mathcal{L}(\mathbf{w}_t)$. To account for the historical magnitude of gradients during training, we define the element-wise accumulated squared gradient Duchi et al. (2011) as:

$$P_t = \sum_{i=1}^{t} \left(\nabla_{\mathbf{w}}\mathcal{L}(\mathbf{w}_i)\right)^2 \tag{2}$$

The inclusion of the logarithmic term $\log_e(P_t + 1)$ introduces a sublinear dampening effect on the learning rate. As training progresses and the accumulated gradient $P_t$ grows, this term increases slowly, ensuring that learning rates decay gradually rather than aggressively. This preserves sufficient learning signal in later iterations, which is particularly beneficial for non-convex landscapes where continued exploration is essential for escaping saddle points or poor local minima Dauphin et al. (2014); Kashyap (2023). The additive constant $\varepsilon > 0$, placed outside the logarithm, serves a distinct purpose: it establishes a lower bound on the denominator, thereby avoiding instability due to division by small values during early training when $P_t$ is close to zero. Importantly, $\varepsilon$ does not interfere with the curvature-based adaptivity introduced by $\log_e(P_t + 1)$, which has been demonstrated in Proposition B.1 that provides superior gradient scaling relative to the square root. Consequently, this formulation preserves gradient-aware scaling while ensuring numerical stability.

This construction yields the following update rule for each parameter dimension:

$$\mathbf{w}_{t+1} = \mathbf{w}_t - \frac{\gamma}{\log_e(P_t + 1) + \varepsilon} \cdot \nabla_{\mathbf{w}}\mathcal{L}(\mathbf{w}_t), \tag{3}$$

**Lemma 3.1.** *Suppose the gradient norm is uniformly bounded by a constant $G > 0$. Then the progress term $P_t$ grows at most linearly with iteration count:*

$$P_t \leq G^2 t.$$

*Proof.* By Accumulated squared gradient's Equation 2, Since $\|\nabla_{\mathbf{w}}\mathcal{L}(\mathbf{w}_i)\| \leq G$ for all $i$, it follows that

$$P_t \leq G^2 t,$$

establishing the claimed linear upper bound. This linear growth ensures the normalization factor in the step size denominator increases gradually but without abrupt escalation, contributing to a stable decay in learning rates. $\square$

**Proposition 3.2.** *Under the assumption that the gradient of the loss function is bounded, i.e.,*

$$\|\nabla_{\mathbf{w}}\mathcal{L}(\mathbf{w}_t)\| \leq G,$$

*the step size in the ZenGrad algorithm is bounded for all $t$. Specifically, for each iteration $t$, the step size $\|\mathbf{w}_{t+1} - \mathbf{w}_t\|$ satisfies the following bound:*

$$\|\mathbf{w}_{t+1} - \mathbf{w}_t\| \leq \frac{\gamma G}{\log_e(P_t + 1) + \varepsilon}.$$

*Proof.* From the update rule,

$$\mathbf{w}_{t+1} = \mathbf{w}_t - \frac{\gamma \nabla_{\mathbf{w}}\mathcal{L}(\mathbf{w}_t)}{\log_e(P_t + 1) + \varepsilon},$$

taking norms and applying the gradient bound yields

$$\|\mathbf{w}_{t+1} - \mathbf{w}_t\| = \frac{\gamma\|\nabla_{\mathbf{w}}\mathcal{L}(\mathbf{w}_t)\|}{\log_e(P_t + 1) + \varepsilon} \leq \frac{\gamma G}{\log_e(P_t + 1) + \varepsilon}.$$

This upper bound explicitly quantifies the maximum possible step length at each iteration, confirming that the update magnitude is effectively regulated by the accumulated gradient information. As $P_t$ grows, the step size shrinks, thus inherently preventing divergence caused by overly large updates. $\square$

**Theorem 3.3** (Lyapunov Stability Sastry (1999) and Convergence of ZenGrad). *Let $L : \mathbb{R}^d \to \mathbb{R}$ be a differentiable objective function with a global minimizer $w^*$, and let $\{w_t\}_{t \geq 0}$ be the sequence of iterates generated by the ZenGrad update rule in Equation 3, where $\gamma > 0$ is the learning rate and $P_t \geq 0$ is an auxiliary term dependent on the gradient history. Assume further that $L$ is $L$-smooth, i.e.,*

$$L(y) \leq L(x) + \nabla L(x)^\top (y - x) + \frac{L}{2}\|y - x\|^2,$$

*and the step-size $\eta_t = \frac{\gamma}{\log_e(P_t+1)+\varepsilon}$ satisfies $\eta_t \leq \frac{1}{L}$ for all $t$. Then, the Lyapunov function*

$$V(w_t) = L(w_t) - L(w^*)$$

*is non-increasing, i.e.,*

$$V(w_{t+1}) \leq V(w_t),$$

*and hence the iterates $w_t$ asymptotically converge towards the global minimum $w^*$ in the sense of objective value.*

*Proof.* To examine the evolution of $V(\mathbf{w}_t)$, we look at the difference between $V(\mathbf{w}_{t+1})$ and $V(\mathbf{w}_t)$:

$$V(\mathbf{w}_{t+1}) - V(\mathbf{w}_t) = (\mathcal{L}(\mathbf{w}_{t+1}) - \mathcal{L}(\mathbf{w}^*)) - (\mathcal{L}(\mathbf{w}_t) - \mathcal{L}(\mathbf{w}^*)).$$

By $L$-smoothness and the update rule $\mathbf{w}_{t+1} = \mathbf{w}_t - \eta_t \nabla \mathcal{L}(\mathbf{w}_t)$, we have

$$\mathcal{L}(\mathbf{w}_{t+1}) \leq \mathcal{L}(\mathbf{w}_t) - \eta_t \|\nabla \mathcal{L}(\mathbf{w}_t)\|^2 + \frac{L}{2}\eta_t^2 \|\nabla \mathcal{L}(\mathbf{w}_t)\|^2.$$

Substituting $\eta_t = \frac{\gamma}{\log_e(P_t+1)+\varepsilon}$, we obtain

$$\mathcal{L}(\mathbf{w}_{t+1}) - \mathcal{L}(\mathbf{w}_t) \leq -\eta_t \left(1 - \frac{L\eta_t}{2}\right) \|\nabla \mathcal{L}(\mathbf{w}_t)\|^2.$$

Since $\eta_t \leq 1/L$, it follows that $1 - \frac{L\eta_t}{2} \geq \frac{1}{2}$, and thus

$$\mathcal{L}(\mathbf{w}_{t+1}) - \mathcal{L}(\mathbf{w}_t) \leq -\frac{\eta_t}{2} \|\nabla \mathcal{L}(\mathbf{w}_t)\|^2.$$

Consequently,

$$V(\mathbf{w}_{t+1}) - V(\mathbf{w}_t) \leq -\frac{\eta_t}{2} \|\nabla \mathcal{L}(\mathbf{w}_t)\|^2 \leq 0,$$

showing $V(\mathbf{w}_{t+1}) \leq V(\mathbf{w}_t)$.

Therefore, the Lyapunov function $V(w_t)$ is non-increasing along the iterates, ensuring Lyapunov stability of the ZenGrad dynamics. Since $V(w_t)$ is bounded below and decreases monotonically, it converges to a finite limit, and $\|\nabla L(w_t)\|^2 \to 0$ as $t \to \infty$. Hence, the iterates $\{w_t\}$ approach a stationary point $w^*$, establishing convergence and stability of the update rule. Further results on nonconvex stationary convergence and global linear convergence under the PL condition are provided in Theorem B.4 and B.5. $\qquad\square$

### 3.2 ZenGrad with Momentum (M-ZenGrad)

While Vanilla ZenGrad achieves adaptive learning by leveraging the accumulated magnitudes of historical gradients, its convergence—especially during the initial phases of training from scratch—can be further accelerated. To address this, we integrate momentum into the ZenGrad framework. In this work, we explore two variants: standard momentum, which follows the conventional formulation employed in stochastic gradient methods (Polyak, 1964), and Nesterov momentum, a widely used extension that anticipates future parameter updates (Nesterov, 1983; Sutskever et al., 2013), leading to improved convergence dynamics. We maintain the element-wise accumulated squared gradient as in Equation 2. The velocity vector with momentum coefficient $\mu \in [0, 1)$ is defined as:

$$\boldsymbol{v}_t = \mu \boldsymbol{v}_{t-1} + \nabla_{\boldsymbol{w}} \mathcal{L}(\boldsymbol{w}_t), \quad \boldsymbol{v}_0 = 0, \quad \boldsymbol{u}_t = \begin{cases} \nabla_{\boldsymbol{w}} \mathcal{L}(\boldsymbol{w}_t) + \mu \boldsymbol{v}_t & \text{(Nesterov)} \\ \boldsymbol{v}_t & \text{(Standard)} \end{cases}$$

$$\boldsymbol{w}_{t+1} = \boldsymbol{w}_t - \frac{\gamma}{\log_e(\boldsymbol{P}_t + 1) + \varepsilon} \cdot \boldsymbol{u}_t \tag{4}$$

## 4 EXPERIMENTS

An experimental framework is designed to evaluate the effectiveness of the proposed optimizers, i.e., ZenGrad and M-ZenGrad, as well as the novel activation function, i.e., LogLU. All experiments are conducted on an NVIDIA RTX A4500 GPU hosted on RunPod, which provides 23.7 TFLOPS using mixed precision (FP16+FP32) for improved computational efficiency. These settings are kept consistent across all evaluations to ensure fair comparisons. Each optimizer is carefully tuned for every task (See Section 5 for hyperparameter tuning details). The experiments cover a variety of standard tasks, considering both training from scratch and using pretrained settings.

### 4.1 IMAGE CLASSIFICATION

We evaluate various datasets and architectures on the image classification benchmarks. We consider ImageNet-1K Russakovsky et al. (2015) and its variants, including ReaL Beyer et al. (2020) and ImageNet-V2 Recht et al. (2019), for large-scale evaluation. CIFAR-10 Krizhevsky (2009) is used to examine performance on smaller-scale datasets. For the ImageNet results, images are processed at the default size of $224^2$ and augmented with random resized crops and horizontal flips, followed by standard normalization. Training uses label smoothing with a factor of 0.1 and automatic mixed precision (AMP).

**Training from Scratch** We train a ResNet-18 model from scratch on the ImageNet-1K dataset for 90 epochs and a ResNet-32 He et al. (2016) model on the CIFAR-10 for 160 epochs, both using a batch size of 256. For CIFAR-10, the learning rate is reduced by a factor of 10 at epochs 80 and 120, while for ImageNet, the learning rate is decayed every 30 epochs by the same factor. On ImageNet-1K, Our proposed method achieved a higher validation accuracy compared to other optimizers, excluding momentum SGD. M-ZenGrad achieves similar to SGD (Polyak, 1964; Robbins, 1951), with a slight increase of +0.07%. However, M-ZenGrad achieves a validation loss of 0.74, significantly lower than the 1.61 obtained with SGD (See Table 5). On CIFAR-10 dataset, most optimizers exhibit similar performance, while ZenGrad is observed to perform more effectively. All results are illustrated in Figure 1.

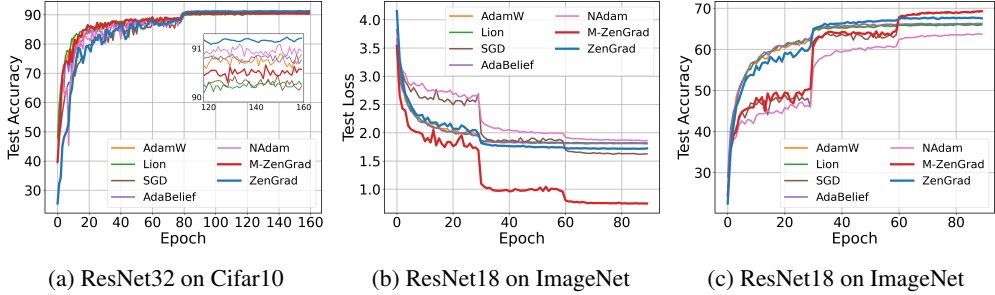

| (a) ResNet32 on Cifar10 | (b) ResNet18 on ImageNet | (c) ResNet18 on ImageNet |
|---|---|---|

Figure 1: Test performance of different optimizers: (a, c) Test accuracy on ResNet32/18 for CIFAR-10 and ImageNet, (b) Test loss on ResNet18 for ImageNet.

**Pre-train on ImageNet-1K** We pretrain the ViT-S/16 Dosovitskiy et al. (2021) model on the ImageNet dataset with a batch size of 256 for 100K steps, employing a cosine annealing scheduler for learning rate decay. Table 1 reports the performance of various optimizers, where standard adaptive methods achieve 70.07–72.35%. The proposed ZenGrad and M-ZenGrad optimizers reach 78.82% and 74.96%, respectively, demonstrating their capability to enhance convergence and performance in large-scale transformer pretraining.

Table 1: Test accuracy ($\mu \pm \sigma$) of multiple optimizers evaluated across different models and datasets.

| Model | Task | AdamW | Lion | NAdam | AdaBelief | ZenGrad | M-ZenGrad |
|---|---|---|---|---|---|---|---|
| ResNet-18 | ImageNet | $66.21 \pm 0.482$ | $66.15 \pm 0.361$ | $63.75 \pm 0.527$ | $66.32 \pm 0.449$ | $67.78 \pm 0.282$ | $\mathbf{69.29 \pm 0.254}$ |
| | Real | $69.45 \pm 0.516$ | $68.51 \pm 0.427$ | $68.46 \pm 0.492$ | $70.28 \pm 0.378$ | $71.31 \pm 0.267$ | $\mathbf{73.23 \pm 0.239}$ |
| | V2 | $54.38 \pm 0.433$ | $54.56 \pm 0.395$ | $52.67 \pm 0.502$ | $55.14 \pm 0.471$ | $55.74 \pm 0.292$ | $\mathbf{57.45 \pm 0.273}$ |
| ViT-S/16 | ImageNet | $70.07 \pm 0.559$ | $72.59 \pm 0.498$ | $72.29 \pm 0.461$ | $72.35 \pm 0.537$ | $\mathbf{78.82 \pm 0.211}$ | $74.96 \pm 0.287$ |
| | Real | $73.24 \pm 0.602$ | $74.80 \pm 0.512$ | $74.54 \pm 0.493$ | $74.21 \pm 0.468$ | $\mathbf{79.60 \pm 0.226}$ | $76.77 \pm 0.294$ |
| | V2 | $60.12 \pm 0.471$ | $61.20 \pm 0.436$ | $61.29 \pm 0.514$ | $61.32 \pm 0.452$ | $\mathbf{67.64 \pm 0.203}$ | $66.84 \pm 0.276$ |

**Transfer Learning** To assess generalization beyond the primary training set, we evaluate ResNet-18 and ViT-S/16 across ImageNet variants, including Real and V2. Table 1 shows that ResNet-18 trained from scratch, M-ZenGrad achieves 69.29% on ImageNet, 73.23% on Real, and 57.45% on V2. For ViT-S/16 (pretrained), ZenGrad reaches 78.82% on ImageNet and 79.60% on Real, with M-ZenGrad demonstrating similarly strong results. On the V2 variant, both proposed optimizers show higher validation metrics than standard adaptive methods. The results indicate consistent improvements across model architectures and dataset variants.

## 4.2 IMAGE SEGMENTATION

We evaluate the Pascal VOC 2012 dataset Everingham et al. (2010) with a U-Net architecture Ronneberger et al. (2015) employing a ResNet-50 encoder under two training protocols: training from scratch for 500 epochs and fine-tuning a pretrained encoder for 200 epochs. All experimental settings were kept fixed, with the optimizer being the only factor varied. Table 2 shows that, the pretrained setting, ZenGrad achieves an IoU of 93.86% and a Dice score of 94.96%, demonstrating its segmentation performance. When trained from scratch, ZenGrad consistently achieves better performance, attaining an IoU of 94.11% and a Dice score of 94.78%, setting it apart from standard adaptive optimizers. Qualitative segmentation results are shown in the images, produced by the ZenGrad model trained from scratch, alongside the corresponding ground-truth annotations (See Figure 6).

Table 2: Evaluation metrics on the Pascal VOC dataset using U-Net with a ResNet-50 encoder, reported as ($\mu \pm \sigma$) across three runs.

| Model | Metric | AdamW | Lion | Adabelief | ZenGrad | M-ZenGrad |
|---|---|---|---|---|---|---|
| | IoU | $90.55 \pm 0.32$ | $91.59 \pm 0.41$ | $91.16 \pm 0.37$ | $\mathbf{93.86 \pm 0.25}$ | $91.81 \pm 0.44$ |
| | Dice | $90.67 \pm 0.28$ | $91.73 \pm 0.35$ | $91.93 \pm 0.33$ | $\mathbf{94.96 \pm 0.21}$ | $90.72 \pm 0.39$ |
| U-Net (ResNet-50) | | | | Training from Scratch | | |
| | IoU | $89.22 \pm 0.36$ | $90.85 \pm 0.28$ | $91.91 \pm 0.30$ | $\mathbf{94.11 \pm 0.22}$ | $90.03 \pm 0.35$ |
| | Dice | $90.34 \pm 0.31$ | $91.61 \pm 0.24$ | $92.63 \pm 0.27$ | $\mathbf{94.78 \pm 0.19}$ | $91.71 \pm 0.29$ |

## 4.3 LANGUAGE MODELING

We conduct experiments on the Wikitext-2 Merity et al. (2016) dataset with vocab size of 50K tokens using a small GPT-style decoder Radford et al. (2018) of 4 transformer layers, 256-dimensional embeddings, 4 self-attention heads, and feed-forward layers with a hidden dimension of $4 \times d_{\text{model}}$. All models are trained with $2^{12}$ tokens per batch for 225K steps. The context length is fixed at 128 tokens, with 0.1 dropout.

Both GeLU and our proposed LogLU activation are employed within the feed-forward layers. For optimizer evaluation, we focus on widely adopted adaptive methods alongside our proposed ZenGrad and M-ZenGrad optimizers, comparing their performance against AdamW and Lion while excluding other optimizers to maintain computational efficiency. Figure 2 shows that, LogLU consistently achieves slightly lower perplexity than GELU, indicating better performance. Among the optimizers, ZenGrad demonstrates the lowest perplexity values, performing better than the other adaptive optimizers and highlighting the advantage of combining it with LogLU.

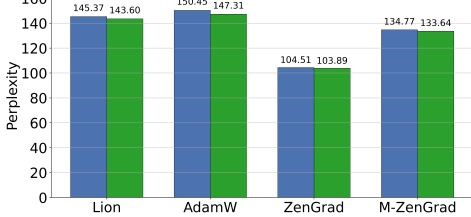

Figure 2: Test PPL on the WikiText-2 dataset using the small GPT-style decoder with different activation functions across optimizers.

## 4.4 ABLATIONS

**Hyperparameter Studies** All experiments are conducted on the ResNet-32 using the CIFAR-10 dataset. First, we analyze the effect of the learning rate, a critical factor for optimization stability and

convergence. We evaluate ZenGrad and M-ZenGrad values using {1e-1, 1e-2, 5e-2, 7e-2, 2e-3, 5e-3}, with AdamW included as a baseline due to its robustness across a wide range of learning rates. Next, we study the role of the epsilon ($\epsilon$) parameter, which prevents division by zero and stabilizes training under low-variance conditions, using values {1e-1, 1e-2, 1e-3, 1e-4, 1e-5, 1e-6, 1e-7, 1e-8}. Finally, we also investigate the momentum-based extension M-ZenGrad, testing momentum coefficients {0.1, 0.3, 0.6, 0.8, 0.9, 0.95, 0.99}, with both standard and Nesterov acceleration evaluated on the same momentum values. The corresponding results are illustrated in Figure 5.

**Effect of Log Variants** We explored the impact of different logarithmic bases on the update rules within the ZenGrad and M-ZenGrad optimization frameworks. While the natural logarithm (base $e$) is commonly used as the default, we also evaluated the use of a logarithm with base 10 to understand its effect on optimization dynamics. These experiments were conducted on the ImageNet -1k using the ResNet-18, trained from scratch for 90 epochs, following the same experimental settings outlined in Section 4.1. The goal was to assess how variations in the logarithmic base influence convergence behavior and generalization performance. As shown in Figure 3, our results indicate that there is no significant difference between using $\log$ base $e$ and $\log$ base 10.

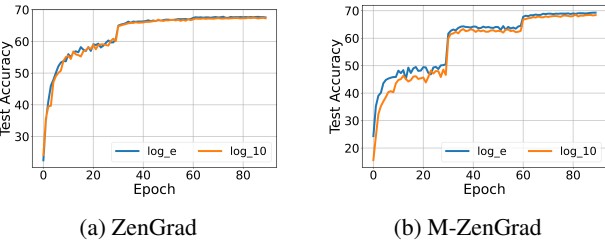

(a) ZenGrad          (b) M-ZenGrad

Figure 3: The impact of logarithmic base on ZenGrad and M-ZenGrad updates is evaluated. ResNet-18 was trained from scratch on ImageNet-1K using $\log_e$ and $\log_{10}$.

**Learning Rate and Weight Decay** We trained the ResNet-18 model on the ImageNet-1K using various combinations of learning rates and weight decay values. All models were trained for 90 epochs with a fixed batch size of 256. We evaluated four optimizer's AdamW, Lion, ZenGrad, and M-ZenGrad, across a grid of learning rates {1e-2, 1e-3, 1e-4} and weight decay values {1e-2, 1e-4, 1e-6}. The results are visualized as heatmaps (See Figure 4), enabling a clear comparison of each optimizer's performance under different regularization settings.

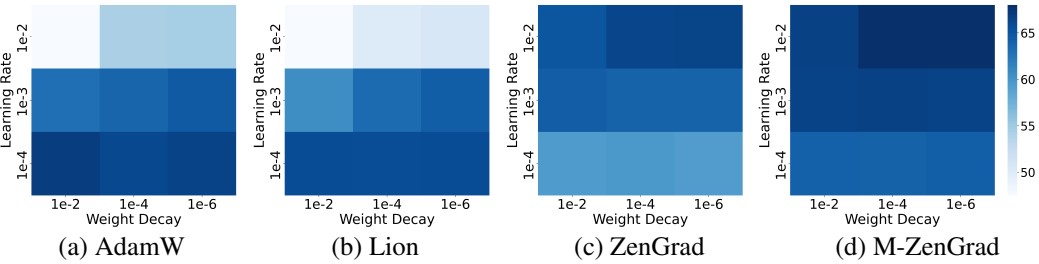

(a) AdamW      (b) Lion      (c) ZenGrad      (d) M-ZenGrad

Figure 4: Ablation study of ResNet-18 on ImageNet-1K under varying learning rate and weight decay configurations across different optimizers.

## 4.5 COMPARISON OF VARIOUS ACTIVATION FUNCTIONS WITH MULTIPLE OPTIMIZERS

To assess the effectiveness of the proposed activation function, experiments were conducted on CIFAR-100 Krizhevsky (2009) using the ResNet-32 architecture for 160 epochs. The training employed a learning rate schedule, where the rate was reduced by a factor of 10 at epochs 80 and 120, with a batch size of 256. Table 3 shows that, LogLU consistently achieved better performance than other activation functions across various optimizers, with most optimizers showing clear gains when paired with it. ZenGrad achieved stronger performance in combination with LogLU, and its momentum-based variant, M-ZenGrad, provided an additional improvement, demonstrating the benefits of pairing effective optimization strategies with well-designed activation functions.

Table 3: Test accuracy ($\mu \pm \sigma$) for different activation functions across various optimizers on CIFAR-100.

| DataSet | A.F | AdamW | Lion | NAdam | AdaBelief | AdaGrad | RMSProp | ZenGrad | M-ZenGrad |
|---|---|---|---|---|---|---|---|---|---|
| CIFAR-100 (ResNet-32) | ReLU | $70.43 \pm 0.41$ | $70.01 \pm 0.44$ | $70.26 \pm 0.39$ | $70.74 \pm 0.42$ | $70.16 \pm 0.40$ | $70.46 \pm 0.43$ | $71.28 \pm 0.26$ | $\mathbf{72.51 \pm 0.19}$ |
| | LeakyReLU | $70.86 \pm 0.38$ | $70.34 \pm 0.41$ | $71.11 \pm 0.43$ | $71.01 \pm 0.40$ | $70.27 \pm 0.39$ | $71.18 \pm 0.42$ | $70.66 \pm 0.24$ | $\mathbf{72.63 \pm 0.18}$ |
| | Swish | $72.22 \pm 0.42$ | $70.30 \pm 0.39$ | $71.23 \pm 0.40$ | $72.07 \pm 0.44$ | $70.62 \pm 0.38$ | $72.31 \pm 0.41$ | $72.33 \pm 0.23$ | $\mathbf{73.29 \pm 0.20}$ |
| | Mish | $71.44 \pm 0.39$ | $70.01 \pm 0.37$ | $72.16 \pm 0.41$ | $71.58 \pm 0.43$ | $70.08 \pm 0.40$ | $71.91 \pm 0.38$ | $70.13 \pm 0.25$ | $\mathbf{73.54 \pm 0.17}$ |
| | GeLU | $70.72 \pm 0.40$ | $70.20 \pm 0.42$ | $71.28 \pm 0.38$ | $71.69 \pm 0.41$ | $70.66 \pm 0.43$ | $71.57 \pm 0.31$ | $72.35 \pm 0.22$ | $\mathbf{73.11 \pm 0.16}$ |
| | Softplus | $71.96 \pm 0.42$ | $71.58 \pm 0.39$ | $73.09 \pm 0.40$ | $72.61 \pm 0.43$ | $70.42 \pm 0.41$ | $72.75 \pm 0.39$ | $72.08 \pm 0.44$ | $\mathbf{73.29 \pm 0.37}$ |
| | LogLU | $72.13 \pm 0.37$ | $72.57 \pm 0.40$ | $72.64 \pm 0.39$ | $72.74 \pm 0.41$ | $72.07 \pm 0.38$ | $72.40 \pm 0.42$ | $72.37 \pm 0.21$ | $\mathbf{73.65 \pm 0.15}$ |

Table 4: Pre-training performance on ImageNet-1K: Test accuracy (%) reported as ($\mu \pm \sigma$) over three runs across optimizers and activation functions.

| Optimizer | ResNet-18 | | ViT/S-16 | |
|---|---|---|---|---|
| | ReLU | LogLU | GELU | LogLU |
| AdamW | $67.42 \pm 0.832$ | $\mathbf{68.68 \pm 0.613}$ | $70.07 \pm 0.559$ | $\mathbf{70.31 \pm 0.447}$ |
| Lion | $67.72 \pm 0.789$ | $\mathbf{68.25 \pm 0.507}$ | $72.59 \pm 0.498$ | $\mathbf{73.04 \pm 0.392}$ |
| ZenGrad | $68.52 \pm 0.593$ | $\mathbf{69.28 \pm 0.482}$ | $78.82 \pm 0.211$ | $\mathbf{79.29 \pm 0.163}$ |
| M-ZenGrad | $70.45 \pm 0.487$ | $\mathbf{71.28 \pm 0.368}$ | $74.96 \pm 0.287$ | $\mathbf{76.29 \pm 0.245}$ |

Furthermore, pre-trained results on ImageNet-1K were obtained after 100K training steps. As reported in Table 4, comparing different optimizers across two architectures (ResNet-18 and ViT/S-16) and activation functions. In both models, the use of LogLU led to more consistent and effective outcomes across all optimizers. ZenGrad and M-ZenGrad showed better performance over AdamW and Lion, especially when combined with LogLU. For ResNet-18, an accuracy of 71.28% was achieved using M-ZenGrad with LogLU, while for ViT/S-16, ZenGrad attained 79.29% with LogLU. These results suggest that integrating LogLU can slightly enhance the performance of the model across different architectures.

## 5 HYPERPARAMETER TUNING

To ensure fair and meaningful comparisons, we systematically tune critical optimization hyperparameters—specifically, the learning rate (`lr`) and decoupled weight decay coefficient ($\lambda$)—across all methods. M-ZenGrad employing a fixed momentum coefficient of $\beta_1 = 0.9$ (See Figure 5 for ablation analysis). Momentum parameters for all optimizers were kept default. The core of ZenGrad lies in its learning rate (See Equation 3). Due to this logarithmic scaling, in our experiments we observed that ZenGrad requires a *5–10x larger learning rate compared to AdamW* to keep the similar intensity. Note that the learning rate value must be adjusted according to the same ratio relative to AdamW, Remaining all other training settings are kept constant throughout the experiments. The optimizer configurations used in all experimental domains—including image classification, segmentation, and language modeling—as:

• $lr = 1e{-}3$, $\lambda = 1e{-}4$ in AdamW; $lr = 1e{-}4$, $\lambda = 1e{-}2$ in Lion; $lr = 1e{-}3$, $\lambda = 1e{-}4$ in NAdam; $lr = 1e{-}3$, $\lambda = 1e{-}8$ in AdaBelief; $lr = 1e{-}2$, $\lambda = 1e{-}4$ in ZenGrad; $lr = 1e{-}2$, $\lambda = 1e{-}4$ in M-ZenGrad.

Hyperparameter tuning is a computationally intensive but essential part of optimizing performance. To better understand the sensitivity of each optimizer, In Figure 4, we present multiple optimizers with various `lr` and $\lambda$ values, trained using ResNet-18 from scratch on the ImageNet. We observe that ZenGrad and M-ZenGrad are more robust, achieving similar performance across a range of hyperparameters compared to AdamW and Lion.

## 6 RELATED WORK

Our work focus on propagation of gradients navigating the complex, non-convex optimization landscapes typical of deep learning. This necessity has driven significant advancements in both optimization algorithms and activation functions. A variety of sophisticated optimizers—AdamW

Loshchilov & Hutter (2019), Lion Chen et al. (2023), AdaBelief Zhuang et al. (2020), AdaGrad Duchi et al. (2011), RMSProp Tieleman (2012), NAdam Dozat (2016), and SGD Robbins (1951) have been engineered to enhance gradient-based training by dynamically adjusting learning rates and stabilizing parameter updates. Alongside these, modern activation functions like ReLU Nair & Hinton (2010), Leaky ReLU Xu et al. (2015), Swish Ramachandran et al. (2017), Mish Misra (2020), and GELU Hendrycks & Gimpel (2023), Softplus Dugas et al. (2000), contribute by introducing nonlinearities that improve gradient stability and model expressiveness. The synergy of these innovations facilitates the effective training of state-of-the-art neural architectures, including Vision Transformers (ViT) Dosovitskiy et al. (2021), ResNets He et al. (2016), and GPT Radford et al. (2018) models, enabling them to capture and learn complex data patterns with greater efficiency.

## 7 CONCLUSION

In this work, we introduced the ZenGrad optimizer, its momentum variant M-ZenGrad, and the LogLU activation function, focusing on improving gradient flow and training stability. Our theoretical analysis confirmed their convergence properties across different types of optimization problems, while extensive experiments demonstrated consistent performance gains across various tasks. Hyperparameter ablations further validated the reliability and adaptability of these methods. These findings highlight the potential of rethinking gradient updates and activation design to achieve more efficient and stable training, offering a foundation for future developments in optimization strategies.

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

## APPENDIX

## A LogLU Proofs

The LogLU activation function introduces an implicit regularization effect in deep neural networks by penalizing large negative pre-activations logarithmically. This effect stabilizes gradient flow and enhances generalization. The LogLU activation function induces sparsity in activations and stabilizes gradient flow by logarithmically penalizing large negative pre-activations. This implicit regularization improves the generalization of deep neural networks.

**Lemma A.1** (Logarithmic Growth of LogLU for $z \leq 0$). *Let the negative branch of the LogLU activation be defined as*

$$f(z) = -\log_e(-z + 1), \quad z \leq 0.$$

*Then, for all $z \leq 0$,*

$$-\log_e(-z + 1) < |z|.$$

*Moreover, the negative branch grows strictly sublinearly with respect to $|z|$:*

$$\lim_{z \to -\infty} \frac{-\log_e(-z + 1)}{|z|} = 0.$$

*Proof.* For any $z \leq 0$, we have $-z + 1 \geq 1$. Since the natural logarithm is strictly increasing and $\log_e(1) = 0$, it follows that $\log_e(-z + 1) \geq 0$, and therefore

$$-\log_e(-z + 1) \leq 0 \leq |z|.$$

Noting that $|z| = -z$ for negative $z$, this immediately establishes the inequality

$$-\log_e(-z + 1) < |z|.$$

To analyze the asymptotic growth, consider the ratio

$$\frac{-\log_e(-z + 1)}{|z|} = \frac{\log_e(-z + 1)}{-z}.$$

As $z \to -\infty$, the logarithmic term grows much more slowly than the linear term $-z$. Consequently,

$$\lim_{z \to -\infty} \frac{\log_e(-z + 1)}{-z} = 0,$$

showing that the negative branch of LogLU grows strictly sublinearly with respect to the magnitude of $z$. This ensures that large negative inputs are penalized gently, reducing the risk of excessively large gradients, improving training stability, while still allowing meaningful negative activations. □

**Corollary A.2.** *The attenuation of gradients for large negative values ensures that excessively negative pre-activations do not dominate the gradient flow, promoting stable optimization dynamics in deep neural networks.*

*Proof.* The regularization effect of LogLU can be understood by analyzing its contribution to the total loss function of a deep neural network. Let $\mathcal{L}_{\text{task}}$ denote the task-specific loss (e.g., cross-entropy or mean squared error). The total loss can be expressed as:

$$\mathcal{L} = \mathcal{L}_{\text{task}} + \lambda \sum_{i: z_i \leq 0} \big[ -\log_e(-z_i + 1) \big], \quad x_i \leq 0$$

where the second term represents an implicit regularization effect introduced by LogLU. The LogLU activation function introduces a logarithmic term, $[-\log_e(-z_i + 1)]$, which effectively discourages large negative activations while minimally impacting small negative values. This property promotes activation sparsity, a desirable characteristic known to enhance generalization in neural networks. Furthermore, the gradient of this penalty diminishes for highly negative $z_i$, inherently stabilizing the gradient flow and preventing issues such as gradient explosion or oscillatory behavior during training. By penalizing large negative pre-activations, LogLU implicitly enforces a constraint on the model's effective capacity, thereby acting as a form of regularization. This regularization mitigates overfitting risks and contributes to improved generalization performance on unseen data. $\square$

## B ZENGRAD PROOFS

**Proposition B.1** (Logarithmic vs. Square Root — Step-size Scaling Inequality)**.** *Let $P_t \geq 0$ and $\varepsilon > 0$. Define the ZenGrad and AdaGrad/Adam denominators as*

$$D_{\text{zengrad}}(P_t) = \log_e(P_t + 1) + \varepsilon, \qquad D_{\text{adagrad}}(P_t) = \sqrt{P_t} + \varepsilon.$$

*Then, for all $P_t \geq 0$,*

$$\boxed{D_{\text{zengrad}}(P_t) \leq D_{\text{adagrad}}(P_t)} \quad \Longrightarrow \quad \boxed{\frac{1}{D_{\text{zengrad}}(P_t)} \geq \frac{1}{D_{\text{adagrad}}(P_t)}},$$

*Consequently, for identical learning rates $\gamma = \eta$,*

$$\Delta_{\text{zengrad}}(P_t) = \frac{\gamma}{\log_e(P_t + 1) + \varepsilon} \geq \frac{\eta}{\sqrt{P_t} + \varepsilon} = \Delta_{\text{adagrad}}(P_t),$$

*indicating that ZenGrad maintains a consistently larger effective step-size compared to Ada-Grad/Adam for all $P_t \geq 0$.*

*Proof.* For all $P_t \geq 0$, which gives $\log_e(P_t + 1) \leq \sqrt{P_t}$ and hence the inequality function is as follows:
$$f(P_t) = \sqrt{P_t} - \log_e(P_t + 1), \quad P_t \geq 0.$$
Compute the derivative for $P_t > 0$:

$$f'(P_t) = \frac{1}{2\sqrt{P_t}} - \frac{1}{1 + P_t}.$$

Note that

$$\frac{1}{2\sqrt{P_t}} - \frac{1}{1 + P_t} \geq 0 \quad \Longleftrightarrow \quad \frac{1}{1 + P_t} \leq \frac{1}{2\sqrt{P_t}} \quad \Longleftrightarrow \quad 1 + P_t \geq 2\sqrt{P_t}.$$

But $1 + P_t - 2\sqrt{P_t} = (\sqrt{P_t} - 1)^2 \geq 0$, so the last inequality holds for all $P_t \geq 0$. Thus $f'(P_t) \geq 0$ for all $P_t > 0$, which means $f$ is nondecreasing on $[0, \infty)$. Since $f(0) = 0$, it follows that $f(P_t) \geq 0$ for every $P_t \geq 0$. Therefore

$$\log_e(P_t + 1) \leq \sqrt{P_t} \qquad (\forall P_t \geq 0).$$

Adding $\varepsilon > 0$ to both sides:
$$\log_e(P_t + 1) + \varepsilon \leq \sqrt{P_t} + \varepsilon,$$

taking reciprocals yields

$$\frac{1}{\log_e(P_t + 1) + \varepsilon} \geq \frac{1}{\sqrt{P_t} + \varepsilon}.$$

The ZenGrad step-size is always greater than or equal to that of AdaGrad/Adam for the same accumulated gradient history. Consequently, ZenGrad's logarithmic scaling yields a slower step-size decay, offering better long-term gradient responsiveness and stability — a desirable property that mitigates the over-damping observed in adaptive gradients. $\square$

---

**Algorithm 1** ZenGrad Optimizer

---

1: **Input:** Objective function $J(\theta)$, initial parameters $\theta_0$, learning rate $\eta$, total steps $T$
2: **Initialize:** $P_0 \leftarrow 0$
3: **for** $t = 1$ **to** $T$ **do**
4:     Compute gradient $g_t \leftarrow \nabla_\theta J(\theta_t)$
5:     Accumulate squared gradients: $P_t \leftarrow P_{t-1} + g_t^2$
6:     Parameter update:

$$\theta_{t+1} \leftarrow \theta_t - \frac{\eta}{\log_e(P_t + 1) + \varepsilon}\, g_t$$

7: **end for**
8: **Return:** $\theta_T$

---

**Proposition B.2.** *If the progress term $P_t$ is monotonically increasing in t, then the effective learning rate*

$$\eta_t = \frac{\gamma}{\log_e(P_t + 1) + \varepsilon}$$

*is a monotonically decreasing function of t. Specifically, for any $t_1 < t_2$,*

$$\eta_{t_1} > \eta_{t_2}.$$

*Proof.* Since $P_t$ is monotonically increasing,

$$P_{t_1} \leq P_{t_2} \implies \log_e(P_{t_1} + 1) \leq \log_e(P_{t_2} + 1).$$

Thus,

$$\frac{1}{\log_e(P_{t_1} + 1) + \varepsilon} > \frac{1}{\log_e(P_{t_2} + 1) + \varepsilon},$$

which immediately implies

$$\eta_{t_1} > \eta_{t_2}.$$

This monotonic decay of the effective learning rate is a desirable property, as it ensures that the optimizer takes progressively smaller steps, facilitating convergence by avoiding oscillations or instability in the later stages of training. $\square$

**Proposition B.3.** *The initial learning rate $\gamma_0$ influences the rate of convergence in ZenGrad. Specifically, if $\gamma_0$ is large, the algorithm will take larger steps initially, leading to faster progress in the early stages of the optimization process. However, as t increases, the progress term $P_t$ causes the learning rate to decay, ensuring stability and fine-tuning of the solution. Conversely, if $\gamma_0$ is small, the algorithm will take smaller steps initially, but still converges effectively as the learning rate decays over time.*

*Proof.* Let the initial learning rate be $\gamma_0$, and consider the learning rate at iteration $t$, which is given by:

$$\eta_t = \frac{\gamma_0}{\log_e(P_t + 1) + \varepsilon}.$$

If $\gamma_0$ is large, the initial updates will be larger, leading to faster progress early on. However, as $t$ increases, the term $\log_e(P_t+1)+\varepsilon$ increases, causing the learning rate to decrease, and the algorithm will settle into a more stable convergence. $\square$

**Theorem B.4** (Convergence in Non-Convex Settings). *Let $\mathcal{L} : \mathbb{R}^d \to \mathbb{R}$ be continuously differentiable and L-smooth:*

$$\mathcal{L}(y) \leq \mathcal{L}(x) + \nabla\mathcal{L}(x)^\top(y - x) + \frac{L}{2}\|y - x\|^2.$$

*Assume $\mathcal{L}$ is bounded below by $\mathcal{L}_{\inf}$. Let*

$$\eta_t = \frac{\gamma}{\log_e(P_t + 1) + \varepsilon}, \quad \gamma/\varepsilon \leq \frac{1}{2L}.$$

---

**Algorithm 2** ZenGrad Optimizer with Momentum (M-ZenGrad)

---

1: **Input:** Objective function $J(\theta)$, initial parameters $\theta_0$, learning rate $\eta$, momentum $\mu$, total steps $T$
2: **Initialize:** $P_0 \leftarrow 0$, $v_0 \leftarrow 0$
3: **for** $t = 1$ **to** $T$ **do**
4:     Compute gradient $g_t \leftarrow \nabla_\theta J(\theta_t)$
5:     Update momentum: $v_t \leftarrow \mu v_{t-1} + g_t$
6:     Define update direction:
$$u_t = \begin{cases} g_t + \mu v_t & \text{(Nesterov)} \\ v_t & \text{(Standard)} \end{cases}$$
7:     Accumulate squared gradients: $P_t \leftarrow P_{t-1} + g_t^2$
8:     Parameter update:
$$\theta_{t+1} \leftarrow \theta_t - \frac{\eta}{\log_e(P_t + 1) + \varepsilon} u_t$$
9: **end for**
10: **Return:** $\theta_T$

---

*Then*

$$\sum_{t=1}^{\infty} \frac{\|g_t\|^2}{\log_e(P_t + 1) + \varepsilon} < \infty, \quad and \quad \lim_{t \to \infty} \|g_t\| = 0.$$

*Hence, every cluster point of $\{w_t\}$ is stationary.*

*Proof.* By $L$-smoothness and $w_{t+1} = w_t - \eta_t g_t$:

$$\mathcal{L}(w_{t+1}) \leq \mathcal{L}(w_t) - \eta_t\Big(1 - \frac{L}{2}\eta_t\Big)\|g_t\|^2 \leq \mathcal{L}(w_t) - \frac{1}{2}\eta_t\|g_t\|^2.$$

Summing over $t$ gives

$$\sum_{t=1}^{T} \eta_t\|g_t\|^2 \leq 2(\mathcal{L}(w_1) - \mathcal{L}_{\inf}) < \infty.$$

Substituting $\eta_t$ yields

$$\sum_{t=1}^{\infty} \frac{\|g_t\|^2}{\log_e(P_t + 1) + \varepsilon} < \infty.$$

If $\|g_t\| \not\to 0$, there exists $c > 0$ and a subsequence $\{t_k\}$ with $\|g_{t_k}\| \geq c$, giving

$$\frac{\|g_{t_k}\|^2}{\log_e(P_{t_k} + 1) + \varepsilon} \gtrsim \frac{c^2}{\log_e k + C},$$

which diverges, a contradiction. Hence $\|g_t\| \to 0$, and continuity of $\nabla\mathcal{L}$ implies all cluster points are stationary. $\qquad\square$

**Theorem B.5** (Global linear convergence under the PL condition). *Assume $\mathcal{L}$ is $L$-smooth and satisfies the Polyak-Łojasiewicz (PL) Karimi et al. (2016) inequality with constant $\mu > 0$:*

$$\|\nabla\mathcal{L}(w)\|^2 \geq 2\mu\big(\mathcal{L}(w) - \mathcal{L}_{\inf}\big) \qquad \text{for all } w.$$

*Let the ZenGrad iterates use step-sizes $\eta_t$ with*

$$0 < \eta_{\min} \leq \eta_t \leq \frac{1}{L} \quad \text{for all } t.$$

*Then the objective decreases geometrically:*

$$\mathcal{L}(w_{t+1}) - \mathcal{L}_{\inf} \leq (1 - \eta_t\mu)\,(\mathcal{L}(w_t) - \mathcal{L}_{\inf}) \leq (1 - \eta_{\min}\mu)^t(\mathcal{L}(w_0) - \mathcal{L}_{\inf}).$$

*In particular, $w_t$ converges linearly in objective value to the global minimum value $\mathcal{L}_{\inf}$.*

*Proof.* From Theorem 3.3 we have

$$\mathcal{L}(w_{t+1}) \leq \mathcal{L}(w_t) - \frac{\eta_t}{2}\|g_t\|^2.$$

Using the PL inequality $\|g_t\|^2 \geq 2\mu(\mathcal{L}(w_t) - \mathcal{L}_{\inf})$ yields

$$\mathcal{L}(w_{t+1}) - \mathcal{L}_{\inf} \leq \mathcal{L}(w_t) - \mathcal{L}_{\inf} - \eta_t\mu(\mathcal{L}(w_t) - \mathcal{L}_{\inf}) = (1 - \eta_t\mu)(\mathcal{L}(w_t) - \mathcal{L}_{\inf}).$$

Since each step-size satisfies $\eta_t \geq \eta_{\min} > 0$, iterating the inequality gives:

$$\mathcal{L}(w_t) - \mathcal{L}_{\inf} \leq (1 - \eta_{\min}\mu)^t(\mathcal{L}(w_0) - \mathcal{L}_{\inf}).$$

Thus, the objective decreases by a constant factor at each step. As a result, the iterates converge linearly to the global minimum $\mathcal{L}_{\inf}$. $\square$

Table 5: Optimizer performance Train from scratch on CIFAR-10 and ImageNet

| DataSet | CIFAR-10 (ResNet-32) | | ImageNet (ResNet-18) | | |
| --- | --- | --- | --- | --- | --- |
| Optimizer | Top-1 | Loss | Top-1 | Loss | Time per Epoch / Memory Usage |
| SGD | 90.51±0.12 | 0.160±0.041 | 69.22±0.32 | 1.61±0.08 | 20.505min / 9.603GB |
| AdamW | 90.89±0.15 | 0.009±0.003 | 66.21±0.48 | 1.81±0.09 | 21.271min / 10.128GB |
| Lion | 90.35±0.11 | 0.005±0.002 | 66.15±0.36 | 1.81±0.08 | 21.172min / 9.712GB |
| NAdam | 91.21±0.17 | 0.009±0.003 | 63.75±0.53 | 1.86±0.09 | 21.313min / 10.304GB |
| Adabelief | 90.96±0.14 | 0.007±0.003 | 66.32±0.45 | 1.80±0.09 | 20.537min / 10.047GB |
| ZenGrad | **91.27±0.09** | **0.004±0.002** | 67.78±0.28 | 1.71±0.07 | 20.454min / 9.695GB |
| M-ZenGrad | 90.66±0.08 | 0.009±0.001 | **69.29±0.25** | **0.74±0.04** | 20.675min / 10.113GB |

**Training Time and Memory Usage.** Across the evaluated optimizers, the per-epoch training time and GPU memory footprint are broadly similar, reflecting comparable computational efficiency. In terms of relative resource requirements, SGD, ZenGrad, and Lion demonstrate the lowest memory consumption, followed by AdamW, AdaBelief, and M-ZenGrad with intermediate usage, while NAdam incurs the highest computational cost. All experiments were conducted on a NVIDIA RTX A4500 GPU hosted on RunPod. Formally, the hierarchy can be expressed as:

$$\text{SGD} \sim \text{ZenGrad} \sim \text{Lion} < \text{AdamW} \sim \text{AdaBelief} \sim \text{M-ZenGrad} < \text{NAdam}.$$

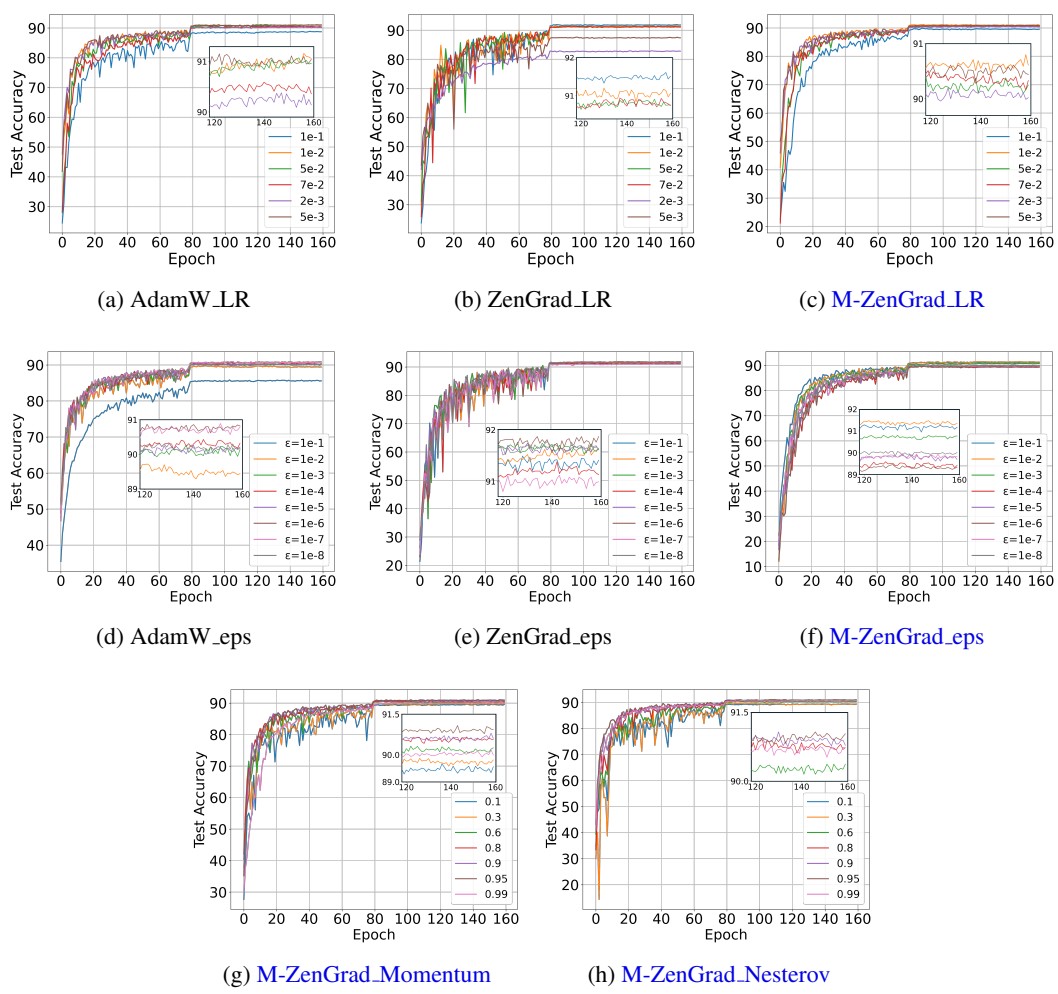

Figure 5: Ablation study on CIFAR-10 with ResNet-32, evaluating the effects of learning rate and $\epsilon$ across AdamW, ZenGrad and M-ZenGrad, and also the impact of standard and Nesterov momentum for M-ZenGrad.

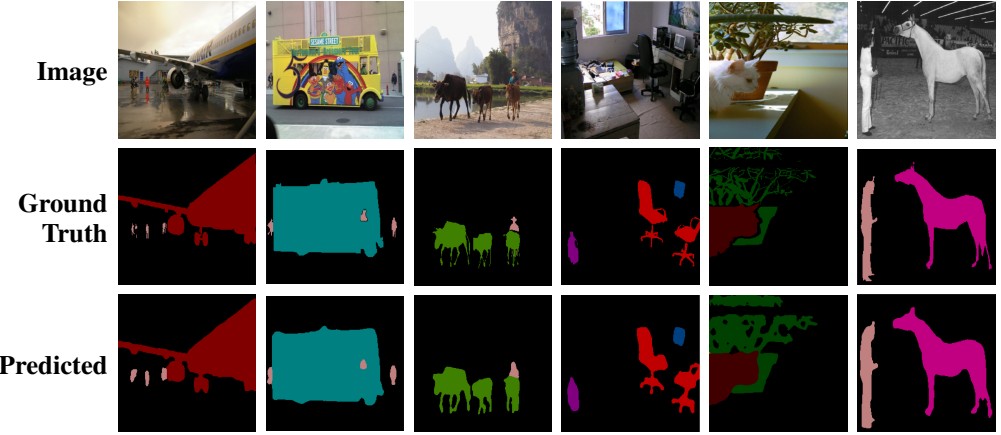

Figure 6: Qualitative segmentation results using ZenGrad on six representative samples. Each column corresponds to a different image. Rows from top to bottom represent: (1) input image, (2) ground truth segmentation mask, and (3) model prediction. The proposed method demonstrates accurate boundary delineation and structural consistency.

