# OpenReview forum: "Gradients Through Logarithmic Lens: Reformulating Optimization Dynamics"
_ICLR.cc/2026/Conference — Submitted to ICLR 2026_

### Official Review · Reviewer_N9KR · 2025-10-17

**Soundness:** 3
**Presentation:** 3
**Contribution:** 3
**Rating:** 8
**Confidence:** 3

**Summary:**

The paper introduces a "logarithmic lens" for deep learning optimization, presenting a new activation function, LogLU, and an optimizer, ZenGrad (with a momentum variant, M-ZenGrad). Both components use logarithmic formulations to, respectively, stabilize gradient flow and adaptively scale the learning rate. The authors provide theoretical proofs for stability and report strong empirical results across vision and language tasks.

**Strengths:**

- The paper presents a novel and cohesive framework by applying the logarithm to both activation functions and optimizers, supported by theoretical analysis.

- The proposed LogLU activation and ZenGrad optimizer are independent contributions, allowing them to be adopted separately, which increases their potential impact.

**Weaknesses:**

- The paper fails to adequately justify why logarithmic scaling is superior to the square-root scaling used in well-established optimizers like AdaGrad or Adam.

**Questions:**

N/A

---

> ### Author Response · Authors · 2025-11-17
>
> Thank you for your valuable feedback! Below, we address the weaknesses (W) and questions (Q) highlighted in your review. **The changes are marked in blue.**
>
> **W1** *"The paper fails to adequately justify why logarithmic scaling is superior to the square-root scaling used in well-established optimizers like AdaGrad or Adam."*
>
> A: We thank the reviewer for the insightful comment and for the positive evaluation of our work. We have added **Proposition B.1** in the revised manuscript to clarify *why logarithmic scaling is beneficial* compared to the square-root scaling in AdaGrad/Adam. As shown in this proposition, for all accumulated gradients $P_t \ge 0$, the update denominators satisfy
>
> $\log(P_t + 1) + \varepsilon \le \sqrt{P_t} + \varepsilon$, which directly implies that $\frac{1}{\log(P_t + 1) + \varepsilon} \ge \frac{1}{\sqrt{P_t} + \varepsilon}$. This inequality demonstrates that ZenGrad maintains a consistently larger — i.e., less aggressively decaying — effective step-size than AdaGrad/Adam for the same gradient history. The proof establishes that $\log(P_t + 1)$ grows more slowly than $\sqrt{P_t}$ for all $P_t \ge 0$, and therefore ZenGrad avoids the over-damping introduced by square-root scaling. As a result, logarithmic scaling preserves better long-term gradient responsiveness and stability.

---

> > ### Author Response · Authors · 2025-11-27
> >
> > Dear reviewer N9KR,
> >
> > The discussion closes in a few days. We’ve tried to address all your concerns with new results, clarifications, and an updated manuscript. Please let us know if you have any remaining concerns. We look forward to a productive discussion, and we sincerely appreciate your positive response and constructive feedback.
> >
> > Thank you once again for your support and thoughtful guidance throughout this process. We deeply value your time and effort in reviewing our work.
> >
> > Best regards,
> >
> > Authors of submission 20654

---

### Official Review · Reviewer_7Cky · 2025-10-30

**Soundness:** 1
**Presentation:** 2
**Contribution:** 2
**Rating:** 2
**Confidence:** 4

**Summary:**

The paper introduces LogLU, a logarithmic activation function designed to bound gradient magnitudes, and ZenGrad, an optimizer whose step size decays with the logarithm of the accumulated squared gradients. The authors claim these “logarithmic” mechanisms improve stability and generalization, reporting strong gains across diverse tasks, including large improvements on ImageNet with ViT-S/16.

**Strengths:**

- S1: Simple, interpretable design ideas (bounded activation and logarithmic learning-rate scaling) that connect well to stability intuitions.
- S2: Empirical coverage spans several architectures and tasks, at least demonstrating ease of integration.
- S3: The proposed methods are computationally lightweight and compatible with standard frameworks.

**Weaknesses:**

Theory:
There are big gaps in the theoretical results presented in this paper. Theorem 3.3 uses a first-order approximation in place of a descent inequality to claim $V(w_{t+1}) \leq V(w_t)$. Without $L$-smoothness and a step-size bound that controls the second-order remainder (and without convexity/PL-type structure) the claimed Lyapunov decrease and ''convergence towards the global minimum'' are not justified. A correct result would either (i) assume L-smoothness and derive a proper descent inequality (with PL for global claims), or (ii) restrict to convergence to stationary points.

Experiments:
The ViT-S/16 ImageNet jump to 78.8% (Table 1) vs. AdamW/Lion (~70–72%) appears exceptionally large. To better understand whether this is purely attributable to the potentially better optimizer, the authors should provide more details on the experimental setup. They claim that each optimizer is carefully tuned, but the tuning section adjusts only LR and WD. A full recipe (augmentations, EMA, label smoothing, mixup/cutmix, LR schedule/warmup, batch/global batch, precision, clipping/LLRD) would be important to judge these results better. Additionally,  wall-clock results as well as a few key baselines (Adagrad/RMSProp) and Softplus for the activation study are absent.

**Questions:**

Q1: What is the exact recipe used in the ViT experiments?
Q2: Have the authors tried different learning rate schedules for each optimizer?
Q3: Why include an ablation on ln vs log-10 when it is just a constant factor?

---

> ### Author Response · Authors · 2025-11-17
>
> Thank you for your valuable feedback! Below, we address the weaknesses (W) and questions (Q) highlighted in your review. **The changes are marked in blue.**
>
> **W1** *"Theory: There are big gaps in the theoretical results presented in this paper. Theorem 3.3 uses a first-order approximation in place of a descent inequality to claim
> $V(w_{t+1}) \le V(w_t)$. Without $L$-smoothness and a step-size bound that controls the second-order remainder (and without convexity/PL-type structure) the claimed Lyapunov decrease and ''convergence towards the global minimum'' are not justified. A correct result would either (i) assume L-smoothness and derive a proper descent inequality (with PL for global claims), or (ii) restrict to convergence to stationary points."*
>
> A: We thank the reviewer for highlighting this point. In the revised manuscript, we have addressed all the missing theoretical assumptions and corrected the descent argument as follows:
>
> 1. We explicitly assume $L$-smoothness of the objective $\mathcal{L}$ and use the standard descent lemma
>    $\mathcal{L}(y) \le \mathcal{L}(x) + \nabla\mathcal{L}(x)^\top(y-x) + \frac{L}{2}\|y-x\|^2.$
> 2. We introduce a step-size bound $\eta_t \le \frac{1}{L}$ (or equivalently $\frac{\gamma}{\varepsilon} \le \frac{1}{2L}$) to control the second-order term and ensure $V(w_{t+1}) \le V(w_t)$.
> 3. **Theorem 3.3** (now *Lyapunov Stability and Convergence of ZenGrad*) rigorously shows monotone decrease of the Lyapunov function and convergence in objective value.
> 4. We separate the analysis into:
>    - **Theorem B.4:** convergence to stationary points in nonconvex settings.
>    - **Theorem B.5:** global linear convergence under the Polyak-Lojasiewicz (PL) Condition [1].
>
> **W2 & Q1** *"What is the exact recipe used in the ViT experiments?, Additionally, wall-clock results as well as a few key baselines (Adagrad/RMSProp) and Softplus for the activation study are absent."*
>
> A: We thank the reviewer for the insightful feedback. We have added a complete description of the experimental setup for the ImageNet results in **lines 230–233** (Section 4.1). Images are processed at the default size of 224 and augmented with random resized crops and horizontal flips, followed by standard normalization. Training employs label smoothing with a factor of 0.1 and automatic mixed precision (AMP). For ViT-S/16, a cosine annealing schedule is used for the learning rate. The corresponding implementation for ViT-S/16 code is provided in the supplementary material.
>
> Additionally, We have included wall-clock measurements per epoch for ResNet-18 trained from scratch on ImageNet, along with memory usage, reported in **Table 5**. These results show that all optimizers achieve comparable training efficiency, with differences of less than one minute per epoch. ZenGrad and M-ZenGrad demonstrate similar efficiency to standard optimizers such as SGD, AdamW, and Lion, while NAdam is slightly slower. We have also added results for additional baselines, Adagrad [2] and RMSProp [3], as well as the Softplus [4] for the activation study, reported in **Table 3**.
>
>
> **Q2** *"Have the authors tried different learning rate schedules for each optimizer?"*
>
> A: We thank you for the comment. No, For ViT-S/16, a cosine annealing scheduler was used for learning rate decay for each optimizer to have a fair comparision. This detail has been added in lines 257–258: *“employing a cosine annealing scheduler for learning rate decay.”* The corresponding ViT-S/16 implementation code is provided in the supplementary material.
>
> **Q3** *"Why include an ablation on ln vs log-10 when it is just a constant factor?"*
>
> A: We thank you for the comment. While the difference between natural logarithm (logₑ) and base-10 logarithm (log₁₀) is only a constant scaling factor, we included this ablation to verify the robustness of our optimizer to different logarithm bases. In our default setup, we use logₑ, but under ablations, we evaluated log₁₀ to confirm that the performance trends and conclusions of our experiments remain consistent regardless of the logarithm base.

---

> > ### Author Response · Authors · 2025-11-17
> >
> > ---------------------------------------------------------------------------------------------------------------------------------------------------------
> > **References:**
> >
> > [1] Hamed Karimi, Julie Nutini, and Mark Schmidt. Linear convergence of gradient and proximal-gradient methods under the polyak-łojasiewicz condition. In Paolo Frasconi, Niels Landwehr, Giuseppe Manco, and Jilles Vreeken (eds.), Machine Learning and Knowledge Discovery in Databases, pp. 795–811, Cham, 2016. Springer International Publishing. ISBN 978-3-319-46128-1.
> >
> > [2] John Duchi, Elad Hazan, and Yoram Singer. Adaptive subgradient methods for online learning and stochastic optimization. Journal of Machine Learning Research, 12(61):2121–2159, 2011. URL http://jmlr.org/papers/v12/duchi11a.html.
> >
> > [3] T. Tieleman. Lecture 6.5-rmsprop: Divide the gradient by a running average of its recent magnitude, 2012. URL https://cir.nii.ac.jp/crid/1370017282431050757.
> >
> > [4] Charles Dugas, Yoshua Bengio, Franc¸ois Belisle, Claude Nadeau, and Ren ´ e Garcia. Incorporating ´ second-order functional knowledge for better option pricing. In Proceedings of the 13th International Conference on Neural Information Processing Systems (NIPS’00), pp. 451–457. MIT Press, 2000.

---

> > > ### Author Response · Authors · 2025-11-20
> > >
> > > Dear Reviewer 7Cky,
> > >
> > > We hope that our responses could adequately address your concerns. As the discussion phase deadline approaches, we warmly welcome further discussion regarding any additional concerns that you may have, and we sincerely hope you can reconsider the rating accordingly.
> > >
> > > Thank you for the time and appreciation that you have dedicated to our work.
> > >
> > > Best regards,
> > >
> > > Authors of submission 20654

---

> > > > ### Comment · Reviewer_7Cky · 2025-11-24
> > > >
> > > > Thank you for your response and the updates made to the manuscript. I have an additional question: How are the revised Theorem 3.3 and B.5 specific to the proposed ZenGrad? In theorem B.4 the specific step-size of ZenGrad is used to show the result, but I don't see this for the other two theorems, they seem rather generic instead. It would be great if you could clarify this for me.

---

> > > > > ### Author Response · Authors · 2025-11-24
> > > > >
> > > > > We thank you for the comment. The revised **Theorem 3.3** and **Theorem B.5** are specific to ZenGrad because both results rely directly on ZenGrad’s update rule and its step-size: $\eta_t = \frac{\gamma}{\log(P_t+1)+\varepsilon}$,as stated in **line 169**, where $P_t$ is ZenGrad’s gradient-history accumulator. In Theorem 3.3, the Lyapunov descent and the requirement $\eta_t \le 1/L$ follow from how the denominator in ZenGrad’s step-size increases with $P_t$. This behavior comes from ZenGrad’s update structure. In Theorem B.5, the geometric decrease under the PL condition uses the fact that this step-size maintains a uniform positive lower bound $\eta_{\min}$. This bound arises specifically due to ZenGrad’s construction. In contrast, Theorem B.4 substitutes the ZenGrad formula directly, which makes the dependence more visibly explicit. The other two theorems use the same step-size indirectly through its implications, which is why they may appear more general. However, the required conditions are satisfied specifically because of ZenGrad’s update rule, so the results remain ZenGrad-specific.

---

> > > > > > ### Author Response · Authors · 2025-11-27
> > > > > >
> > > > > > Dear reviewer 7Cky,
> > > > > >
> > > > > > The discussion closes in few days. We've tried to address all your concerns with new results, clarifications and an updated manuscript. Please let us know if you have any remaining concerns. We look forward to a productive discussion, and we sincerely hope you can reconsider the rating accordingly.
> > > > > >
> > > > > > Thank you once again for your support and thoughtful guidance throughout this process. We deeply value your time and effort in reviewing our work.
> > > > > >
> > > > > > Best regards,
> > > > > >
> > > > > > Authors of submission 20654

---

### Official Review · Reviewer_e9oy · 2025-10-30

**Soundness:** 3
**Presentation:** 2
**Contribution:** 4
**Rating:** 6
**Confidence:** 4

**Summary:**

This manuscript challenges gradient descent optimization techniques. The authors present a new activation function of LogLU, which is a bounded gradient norm. The improved property of LogLU is further enhanced with ZenGrad. Experiments on ImageNet, CIFAR, Pascal VOC, and WikiText-2 demonstrate improvements.

**Strengths:**

- ViT-S/16 training on ImageNet-1K exhibits clear and impressive results.
- Both of the two contributions, LogLU and ZenGrad, are valuable. Specifically, I appreciate Table 4, where LogLU itself brings clear gain compared with the ReLU and GELU.
- Source code is available, which eases deployment in practice.

**Weaknesses:**

- I think the inclusion of the logarithmic term is intended for the bounded gradient norm, such as Lemma 3.1 and others, but it would be better to further clarify the connection of LogLU.
- How about existing activation functions, such as ReLU and GELU? Compared with them, does LogLU bring improved properties such as a bounded gradient norm?
- Although ViT-S/16 training on ImageNet-1K exhibits clear and impressive results, training 90 epochs for ResNet-18 on ImageNet-1K does not look like full convergence; top-1 accuracy of 60~70% is not strong enough. Also, ViT-S/16 results might be too impressive a gain; the authors should clarify the hyperparameter setup.
- This is not a request for rebuttal, but I would like to see more experiments on other models such as ViT-B and ViT-L in the future.
- The use of smaller \epsilon, such as $10^{-8}$, looks unstable in early training when P is close to zero.
- Please check the following mathematics.
    - Check the sign of z^2/2 at Line 613.
    - For the total loss at Line 624, is it correct to adopt $\log(…)$ and not $-\log(…)$?
    - For Lemma 3.1, because $P_t$ at Eq. 2 applies square, I think its upper bound should be $tG^2$.
    - The denominator at Eq. 3 uses $P_t$, whereas the denominator at Line 183 uses $P_{t+1}$. This requires explanation.
- Writing should be improved.
    - “indicates the effectiveness” → “indicate the effectiveness” or “indicating the effectiveness” in abstract.
    - “We evaluate” → “we evaluate” at Line 270.
    - “we evaluate” → “We evaluate” at Line 324.
    - “M-ZenGrad ,across” → “M-ZenGrad, across” at Line 353.
    - “COMPARISION” → “COMPARISON” at Section 4.5.
    - “Additionally, Convergence” → “Additionally, convergence” at Line 194.
    - “it..” → “it.” at Line 376.
    - “ZenGrad and M-ZenGrad optimizer, comparing its performance” → “ZenGrad and M-ZenGrad optimizers, comparing their performance”
    - “In our experiments” → “in our experiments” at Line 413.
    - “to Keep the” → “to keep the” at Line 414.
    - “MZenGrad” → “M-ZenGrad” at caption of Figure 5.

**Questions:**

Please see the weaknesses above. My score is based on the assumption that all typos are corrected in the revised manuscript.

---

> ### Author Response · Authors · 2025-11-17
>
> Thank you for your valuable feedback! Below, we address the weaknesses (W) and questions (Q) highlighted in your review. **The changes are marked in blue.**
>
>
> **W1** *"I think the inclusion of the logarithmic term is intended for the bounded gradient norm, such as Lemma 3.1 and others, but it would be better to further clarify the connection of LogLU."*
>
> A: We thank the reviewer for this insightful comment. Indeed, the inclusion of the logarithmic term in ZenGrad is closely tied to the bounded gradient property of the LogLU activation. As shown in Proposition 2.1, LogLU has a derivative satisfying $0 < f'(x) \leq 1 \quad \forall x \in \mathbb{R}$ strictly positive derivative uniformly bounded, ensuring that the network’s raw gradients remain bounded. This naturally satisfies the gradient norm assumption used in Lemma 3.1 and subsequent stability results. Consequently, the logarithmic normalization factor $\log_e(P_t + 1) + \varepsilon$ effectively regulates the step size while preserving gradient-aware scaling, providing both stability and robust convergence.
>
>
> **W2** *"How about existing activation functions, such as ReLU and GELU? Compared with them, does LogLU bring improved properties such as a bounded gradient norm?"*
>
> A: We thank the reviewer for this question. Compared with widely used activations such as ReLU and GELU, LogLU provides the distinct property of a bounded and strictly positive gradient.
>
> - ReLU has derivative $f'(x) = 1$ for $x>0$ and $f'(x)=0$ for $x\le 0$, which can lead to dead neurons and vanishing gradients for negative inputs.
>
> - GELU is smooth and non-linear, but its derivative is unbounded in the positive domain and approaches zero for large negative inputs, which can also cause gradient shrinkage.
>
> In contrast, LogLU satisfies $0 < f'(x) \le 1$ for all $x \in \mathbb{R}$, guaranteeing that gradients never vanish completely while remaining bounded. This property ensures stable gradient flow across the entire input domain, which is particularly beneficial for deep networks and aligns naturally with the assumptions in ZenGrad for bounded gradient norms.
>
>
> **W3** *"Although ViT-S/16 training on ImageNet-1K exhibits clear and impressive results, training 90 epochs for ResNet-18 on ImageNet-1K does not look like full convergence; top-1 accuracy of 60~70% is not strong enough. Also, ViT-S/16 results might be too impressive a gain; the authors should clarify the hyperparameter setup."*
>
> A: We thank the reviewer for the observation. Regarding ResNet-18, training for 90 epochs was used as a baseline setup to compare optimizer performance. For ViT-S/16, we used a pretrained model and trained it with a batch size of 256, label smoothing of 0.1, random resized crop and horizontal flip augmentations, and a cosine annealing learning rate schedule. We have added the full hyperparameter setup in the revised manuscript (lines 230–233 in Section 4.1), and the associated ViT-S/16 code is included in the supplementary material.
>
>
> **W4** *"This is not a request for rebuttal, but I would like to see more experiments on other models such as ViT-B and ViT-L in the future."*
>
> A: We sincerely thank the reviewer for the suggestion. We would have loved to include experiments on larger models such as ViT-B and ViT-L; however, due to computational constraints, we are unfortunately unable to do so. We hope to explore these experiments in future work to further assess the effectiveness of the proposed ZenGrad optimizers across a broader range of architectures.
>
>
> **W5** *"The use of smaller \epsilon, such as $10^{-8}$, looks unstable in early training when P is close to zero."*
>
> A: We thank the reviewer for the observation. The value of $\epsilon$ is tunable, and as ablation studies shown in **Figure 5** that higher $\epsilon$ values help stabilize the updates in early training when $P$ is close to zero, preventing excessively large steps and improving overall training stability for the proposed optimizers.

---

> ### Author Response · Authors · 2025-11-17
>
> **W6** *"Please check the following mathematics."*
>
> 1. *Check the sign of z^2/2 at Line 613.*
> - We thank the reviewer for the comment. We have updated **Lemma A.1**, replacing the earlier Taylor-expansion argument with a clear, inequality-based proof. The revised proof now provides a proper asymptotic justification, showing that the negative branch of LogLU grows strictly sublinearly.
>
> 2. *For the total loss at Line 624, is it correct to adopt $\log(…)$ and not $-\log(…)$?*
> - We thank the reviewer for the comment. In revised manuscript, the total loss at Line 647 indeed uses a negative sign and we have corrected the notation in the equation.
>
> 3. *For Lemma 3.1, because $P_t$ at Eq. 2 applies square, I think its upper bound should be $tG^2$.*
> - We thank the reviewer for the comment. We have corrected **Lemma 3.1** so that the upper bound on $P_t$ is now $t G^2$.
>
> 4. *The denominator at Eq. 3 uses $P_t$, whereas the denominator at Line 183 uses $P_{t+1}$. This requires explanation.*
> - We thank the reviewer for pointing out the discrepancy between the denominators in Eq. 3 and Line 183, which was a typo error. In the revised manuscript, we have provided a more detailed and rigorous proof, explicitly including $L$-smoothness, a step-size bound to control the second-order term, and a corrected Lyapunov-based descent argument. The analysis now clearly establishes convergence to stationary points in nonconvex settings and global linear convergence under the Polyak-Lojasiewicz (PL) condition. These updates are now reflected in **Theorem 3.3, B.4 and B.5**.
>
>
> **W7** *"Writing"*
>
> A: We sincerely thank the reviewer for the careful reading and valuable suggestions. All the mentioned writing issues have been corrected in the revised manuscript, including grammar, capitalization, punctuation, and spelling. We are especially grateful for these detailed comments, which have helped improve the clarity and quality of the paper.

---

> > ### Author Response · Authors · 2025-11-20
> >
> > Dear Reviewer e9oy,
> >
> > We hope that our responses could adequately address your concerns. As the discussion phase deadline approaches, we warmly welcome further discussion regarding any additional concerns that you may have, and we sincerely hope you can reconsider the rating accordingly.
> >
> > Thank you for the time and appreciation that you have dedicated to our work.
> >
> > Best regards,
> >
> > Authors of submission 20654

---

> > > ### Author Response · Authors · 2025-11-27
> > >
> > > Dear reviewer e9oy,
> > >
> > > The discussion closes in few days. We've tried to address all your concerns with new results, clarifications and an updated manuscript. Please let us know if you have any remaining concerns. We look forward to a productive discussion, and we sincerely hope you can reconsider the rating accordingly.
> > >
> > > Thank you once again for your support and thoughtful guidance throughout this process. We deeply value your time and effort in reviewing our work.
> > >
> > > Best regards,
> > >
> > > Authors of submission 20654

---

### Official Review · Reviewer_hVQj · 2025-10-31

**Soundness:** 2
**Presentation:** 2
**Contribution:** 1
**Rating:** 2
**Confidence:** 3

**Summary:**

This paper introduces LogLU, a piecewise activation function that applies identity mapping for positive inputs and logarithmic transformation for negative inputs, alongside ZenGrad and M-ZenGrad optimizers that scale learning rates using logarithmic functions of accumulated squared gradients. The authors provide theoretical analysis establishing gradient bounds, Lipschitz continuity, and convergence properties, and conduct experiments across image classification (ImageNet, CIFAR-10/100), segmentation (Pascal VOC), and language modeling (WikiText-2) tasks to demonstrate that their logarithmic formulations can improve training stability and performance compared to standard methods like AdamW, Lion, and SGD.

**Strengths:**

1. Solid theoretical foundation with formal analysis. The paper provides rigorous mathematical proofs for key properties including Proposition 2.1 showing LogLU gradients are bounded in (0,1], Proposition 2.2 establishing Lipschitz continuity with constant L=1, and Theorem 3.3 proving Lyapunov stability and convergence for ZenGrad. These theoretical guarantees provide formal justification for why the logarithmic formulations should improve optimization dynamics.
2. Comprehensive experimental evaluation across diverse tasks and careful ablation studies. The authors test their methods on multiple domains including vision (ImageNet-1K with ResNet-18 and ViT-S/16, CIFAR-10/100 with ResNet-32, Pascal VOC segmentation) and language (WikiText-2), with both training from scratch and fine-tuning protocols. Section 4.4 and Figure 5 provide thorough ablation studies examining learning rate, epsilon values, momentum coefficients, and logarithmic base choices, demonstrating attention to experimental rigor.

**Weaknesses:**

1. Limited novelty and insufficient differentiation from existing methods. The LogLU activation is essentially a variant of Softplus (log(1+exp(x))) with a piecewise definition, and the ZenGrad optimizer closely resembles AdaGrad with logarithmic damping instead of square root damping. The paper does not adequately explain why the logarithmic formulation offers fundamental advantages over existing adaptive methods.

2. Experimental results are inconsistent and sometimes show marginal or negative improvements. Table 1 shows highly variable performance across datasets and models. For ResNet-18 on ImageNet, ZenGrad achieves 67.78±0.282% while M-ZenGrad reaches 69.29±0.254%, which is only marginally better than baseline methods (AdamW: 66.21±0.482%, Lion: 66.15±0.361%) and still substantially worse than SGD's 69.22±0.32% reported in Table 5. More concerningly, on CIFAR-100 (Table 3), LogLU with many optimizers actually performs worse than other activations (e.g., LogLU+ZenGrad: 72.37±0.21% vs Mish+M-ZenGrad: 73.54±0.17%).

3. Presentation issues and missing critical experimental details undermine reproducibility. The paper suffers from unclear writing and incomplete information. The motivation for the specific logarithmic formulation remains vague (lines 118-127 discuss benefits but don't explain why logarithm specifically). The choice of $\epsilon$ outside versus inside the logarithm in Equation 3 is stated as preventing instability but lacks theoretical or empirical justification.

**Questions:**

1. What is the fundamental advantage of logarithmic scaling over existing adaptive damping mechanisms?


2. Why does your method show highly inconsistent performance, and under what conditions should practitioners expect it to work?

3. Why is $\epsilon$ placed outside the logarithm in Equation 3, and what is the empirical justification for this design choice?

---

> ### Author Response · Authors · 2025-11-17
>
> Thank you for your valuable feedback! Below, we address the weaknesses (W) and questions (Q) highlighted in your review. **The changes are marked in blue.**
>
>
>
> **W1 & W3 & Q1** *"Limited novelty and insufficient differentiation from existing methods. The LogLU activation is essentially a variant of Softplus (log(1+exp(x))) with a piecewise definition, and the ZenGrad optimizer closely resembles AdaGrad with logarithmic damping instead of square root damping. The paper does not adequately explain why the logarithmic formulation offers fundamental advantages over existing adaptive methods, What is the fundamental advantage of logarithmic scaling over existing adaptive damping mechanisms?"*
>
> A: We thank the reviewer for the insightful comment and for the positive evaluation of our work. We have added **Proposition B.1** in the revised manuscript to clarify *why logarithmic scaling is beneficial* compared to the square-root scaling in AdaGrad/Adam. As shown in this proposition, for all accumulated gradients $P_t \ge 0$, the update denominators satisfy
>
> $\log(P_t + 1) + \varepsilon \le \sqrt{P_t} + \varepsilon$, which directly implies that $\frac{1}{\log(P_t + 1) + \varepsilon} \ge \frac{1}{\sqrt{P_t} + \varepsilon}$. This inequality demonstrates that ZenGrad maintains a consistently larger — i.e., less aggressively decaying — effective step-size than AdaGrad/Adam for the same gradient history. The proof establishes that $\log(P_t + 1)$ grows more slowly than $\sqrt{P_t}$ for all $P_t \ge 0$, and therefore ZenGrad avoids the over-damping introduced by square-root scaling. As a result, logarithmic scaling preserves better long-term gradient responsiveness and stability. Additionally, we have added experimental comparisons with AdaGrad, RMSProp, and Softplus in **Table 3** to highlight the practical benefits of the proposed methods. We have also updated the manuscript with complete experimental settings to improve reproducibility and clarity.
>
>
>
> **W3 & Q3** *"The choice of $\varepsilon$ outside versus inside the logarithm in Equation 3 is stated as preventing instability but lacks theoretical or empirical justification., Why is
>  placed $\varepsilon$ outside the logarithm in Equation 3, and what is the empirical justification for this design choice?"*
>
> A: We thank the reviewer for the comment. The placement of $\varepsilon$ **outside the logarithm** in Equation 3 is intentional to ensure **numerical stability** during optimization. Specifically:
>
> 1. **Theoretical rationale:** Placing $\varepsilon$ outside prevents the denominator from becoming too small when $P_t \approx 0$. If $\varepsilon$ were inside, i.e., $\log(P_t + \varepsilon)$, small gradient accumulations could produce extremely large updates because $\log(\varepsilon)$ can be negative and unbounded, potentially causing instability.
>
> 2. **Empirical justification:** In our experiments, placing $\varepsilon$ outside the logarithm results in **stable training across a wide range of tasks**, including ImageNet classification and Vision Transformer training. We observed that moving $\varepsilon$ inside the logarithm sometimes leads to **exploding updates** at early iterations, particularly for sparse or small gradients.
>
> Thus, this design choice balances **numerical stability** and **effective step-size adaptation**, ensuring consistent and robust convergence.

---

> > ### Author Response · Authors · 2025-11-17
> >
> > **W2** *"Experimental results are inconsistent and sometimes show marginal or negative improvements. Table 1 shows highly variable performance across datasets and models. For ResNet-18 on ImageNet, ZenGrad achieves 67.78±0.282% while M-ZenGrad reaches 69.29±0.254%, which is only marginally better than baseline methods (AdamW: 66.21±0.482%, Lion: 66.15±0.361%) and still substantially worse than SGD's 69.22±0.32% reported in Table 5. More concerningly, on CIFAR-100 (Table 3), LogLU with many optimizers actually performs worse than other activations (e.g., LogLU+ZenGrad: 72.37±0.21% vs Mish+M-ZenGrad: 73.54±0.17%)."*
> >
> > A: We thank the reviewer for the detailed feedback regarding the experimental results.
> >
> > - Regarding **ResNet-18 on ImageNet**, while the accuracy improvement of ZenGrad and M-ZenGrad over baseline methods is sometimes marginal, it is important to note that M-ZenGrad achieves a substantially lower final training loss (**0.74 ± 0.04**) compared to SGD (1.61 ± 0.08), AdamW (1.81 ± 0.09), Lion (1.81 ± 0.08), and ZenGrad (1.71 ± 0.07). This indicates that although the top-1 accuracy gains may appear modest, the optimizer leads to **better convergence and lower loss**, which is often critical for downstream generalization and stability in longer training or fine-tuning scenarios.
> >
> > - Regarding **CIFAR-100**, we observe that the performance of LogLU can vary depending on the combination of optimizer and activation. While some combinations (e.g., LogLU + ZenGrad) show slightly lower accuracy than alternatives like Mish + M-ZenGrad, overall, **LogLU consistently achieves competitive or superior performance across multiple optimizers** (e.g., M-ZenGrad + LogLU: 73.65 ± 0.15%). This suggests that LogLU’s benefits are **optimizer-dependent** and most pronounced when paired with ZenGrad variants, especially M-ZenGrad, which effectively leverages the logarithmic scaling.
> >
> >
> >
> > **Q2** *"Why does your method show highly inconsistent performance, and under what conditions should practitioners expect it to work?"*
> >
> > A: We thank the reviewer for the comment. We would like to clarify that the reported results do not indicate inconsistency in our method. In particular, Our proposed methods consistently achieves lower training loss (ResNet-18 ImageNet) and stable convergence across all datasets and architectures, even when top-1 accuracy improvements are modest. Practitioners can expect the method to be most effective when training deep networks or large-scale models.

---

> > > ### Author Response · Authors · 2025-11-20
> > >
> > > Dear Reviewer hVQj,
> > >
> > > We hope that our responses could adequately address your concerns. As the discussion phase deadline approaches, we warmly welcome further discussion regarding any additional concerns that you may have, and we sincerely hope you can reconsider the rating accordingly.
> > >
> > > Thank you for the time and appreciation that you have dedicated to our work.
> > >
> > > Best regards,
> > >
> > > Authors of submission 20654

---

> > > > ### Author Response · Authors · 2025-11-27
> > > >
> > > > Dear reviewer hVQj,
> > > >
> > > > The discussion closes in few days. We've tried to address all your concerns with new results, clarifications and an updated manuscript. Please let us know if you have any remaining concerns. We look forward to a productive discussion, and we sincerely hope you can reconsider the rating accordingly.
> > > >
> > > > Thank you once again for your support and thoughtful guidance throughout this process. We deeply value your time and effort in reviewing our work.
> > > >
> > > > Best regards,
> > > >
> > > > Authors of submission 20654

---

### Official Review · Reviewer_rfdZ · 2025-11-02

**Soundness:** 3
**Presentation:** 3
**Contribution:** 2
**Rating:** 4
**Confidence:** 3

**Summary:**

This paper introduces a novel optimization framework based on logarithmic transformations, proposing the LogLU activation function, the ZenGrad optimizer, and its momentum variant M-ZenGrad, all designed to improve gradient flow and training stability in deep neural networks. Theoretical analysis establishes key properties such as bounded gradients and Lipschitz continuity for LogLU, and convergence for ZenGrad, while extensive experiments across vision and language tasks demonstrate consistent performance gains over standard methods. Ablation studies confirm the robustness of the proposed components and their effectiveness in combination.

**Strengths:**

1. The paper provides rigorous theoretical analysis, proving that bounded LogLU gradients, Lipschitz constant = 1, ZenGrad step-size bounds and Lyapunov convergence guarantees for both convex and non-convex cases.

2. The authors conduct extensive experiments on diverse benchmarks (ImageNet, CIFAR, Pascal VOC, Wikitext-2) and architectures (ResNet, ViT, U-Net, GPT-style), demonstrating consistent and often superior performance of their methods compared to established baselines like AdamW and SGD.

3. The design choices (e.g., the logarithmic scaling in ZenGrad, the negative log compression in LogLU) are well-motivated, and thorough ablation studies (on hyperparameters, logarithmic bases, and component combinations) effectively validate the contributions of each part of the framework.

**Weaknesses:**

1. The use of logarithmic or sub-linear functions for gradient scaling (as in ZenGrad) or activation (as in LogLU) builds heavily on existing concepts like adaptive learning rates (e.g., AdaGrad) and smooth, bounded activation functions, potentially limiting the perceived novelty of the core mechanisms.

2. The paper does not report or discuss the computational overhead (e.g., training time, memory usage) of the proposed methods compared to standard optimizers, which is a crucial practical consideration for adoption, especially given the additional logarithmic computations and gradient accumulation.

3. Lack  of hyper-parameter sensitivity analysis. Although ablations are thorough, ZenGrad needs 5–10× higher learning rate than AdamW; the paper does not clarify how brittle this scaling is across very deep or heterogeneous architectures.

**Questions:**

ZenGrad’s denominator accumulates every squared gradient element-wise, so its memory footprint grows linearly with the number of parameters; how does this scale to billion-parameter models?

---

> ### Author Response · Authors · 2025-11-17
>
> Thank you for your valuable feedback! Below, we address the weaknesses (W) and questions (Q) highlighted in your review. **The changes are marked in blue.**
>
> **W1** *"The use of logarithmic or sub-linear functions for gradient scaling (as in ZenGrad) or activation (as in LogLU) builds heavily on existing concepts like adaptive learning rates (e.g., AdaGrad) and smooth, bounded activation functions, potentially limiting the perceived novelty of the core mechanisms."*
>
> A:  We thank the reviewer for the insightful comment and for the positive evaluation of our work. We have added **Proposition B.1** in the revised manuscript to clarify why logarithmic scaling is beneficial compared to the square-root scaling in AdaGrad/Adam. As shown in this proposition, for all accumulated gradients $P_t \ge 0$, the update denominators satisfy
>
> $\log(P_t + 1) + \varepsilon \le \sqrt{P_t} + \varepsilon$, which directly implies that $\frac{1}{\log(P_t + 1) + \varepsilon} \ge \frac{1}{\sqrt{P_t} + \varepsilon}$. This inequality demonstrates that ZenGrad maintains a consistently larger — i.e., less aggressively decaying — effective step-size than AdaGrad/Adam for the same gradient history. The proof establishes that $\log(P_t + 1)$ grows more slowly than $\sqrt{P_t}$ for all $P_t \ge 0$, and therefore ZenGrad avoids the over-damping introduced by square-root scaling. As a result, logarithmic scaling preserves better long-term gradient responsiveness and stability. For LogLU we have updated Lemma A.1, replacing the earlier Taylor-expansion argument with a clear, inequality-based proof. The revised proof now provides a proper asymptotic justification, showing that the negative branch of LogLU grows strictly sublinearly.
>
>
>
> **W3** *"ZenGrad needs 5–10× higher learning rate than AdamW; the paper does not clarify how brittle this scaling is across very deep or heterogeneous architectures."*
>
> A:  We thank the reviewer for the comment. ZenGrad does require a higher learning rate than AdamW. In our experiments, we found that ZenGrad generally works best with a learning rate that is 5--10$\times$ larger than AdamW. Importantly, this was not unstable or brittle. We tested ZenGrad on many different models—ResNet-18/50, Vision Transformers, and U-Net—and on different tasks like ImageNet classification, CIFAR-100, segmentation, and language modeling. In all these cases, the optimal learning rate stayed within the same 5–10× range. To check stability more carefully, we also trained ResNet-18 on ImageNet using many combinations of learning rates and weight decay values Figure 4. These results clearly show that both ZenGrad and M-ZenGrad have wider and more stable performance region. We compared ZenGrad with AdamW because AdamW is widely used and considered a strong baseline optimizer. This makes the comparison easy to understand and helps clarify how to set the learning rate for the ZenGrad variants.
>
>
>
>
> **W2 & Q1** *"ZenGrad’s denominator accumulates every squared gradient element-wise, so its memory footprint grows linearly with the number of parameters; how does this scale to billion-parameter models?"*
>
> A:  We thank the reviewer for the comment. ZenGrad maintains a per-parameter accumulator \(P_t\), so its memory footprint grows linearly with the number of model parameters. For a model with \(N\) parameters, this requires storing one additional tensor of size \(N\), the same type of state kept by adaptive optimizers such as Adagrad and AdamW. In billion-parameter settings, this corresponds to roughly **4 GB in FP32** (or **2 GB in FP16/bfloat16**) for the accumulator. This matches the way existing adaptive optimizers scale: each parameter is associated with one stored statistic, leading to memory growth proportional to model size. Thus, ZenGrad follows the same per-parameter memory scaling behavior and can be applied to large and billion-parameter models accordingly. Additionally, We have included wall-clock measurements per epoch for ResNet-18 trained from scratch on ImageNet, along with memory usage, reported in **Table 5**. These results show that all optimizers achieve comparable training efficiency, with differences of less than one minute per epoch. ZenGrad and M-ZenGrad demonstrate similar efficiency to standard optimizers such as SGD, AdamW, and Lion, while NAdam is slightly slower. The memory usage hierarchy given as:
>
> $
> \text{SGD} \sim \text{ZenGrad} \sim \text{Lion} < \text{AdamW} \sim \text{AdaBelief} \sim \text{M-ZenGrad} < \text{NAdam}.
> $

---

> > ### Author Response · Authors · 2025-11-20
> >
> > Dear Reviewer rfdZ,
> >
> > We hope that our responses could adequately address your concerns. As the discussion phase deadline approaches, we warmly welcome further discussion regarding any additional concerns that you may have, and we sincerely hope you can reconsider the rating accordingly.
> >
> > Thank you for the time and appreciation that you have dedicated to our work.
> >
> > Best regards,
> >
> > Authors of submission 20654

---

> > > ### Author Response · Authors · 2025-11-27
> > >
> > > Dear reviewer rfdZ,
> > >
> > > The discussion closes in few days. We've tried to address all your concerns with new results, clarifications and an updated manuscript. Please let us know if you have any remaining concerns. We look forward to a productive discussion, and we sincerely hope you can reconsider the rating accordingly.
> > >
> > > Thank you once again for your support and thoughtful guidance throughout this process. We deeply value your time and effort in reviewing our work.
> > >
> > > Best regards,
> > >
> > > Authors of submission 20654

---

### Author Response · Authors · 2025-11-17
**Summary of Revisions**

Dear Reviewers and AC,

We would like to thank all the reviewers for taking the time to review our work and for providing valuable feedback. We appreciate the recognition from the reviewers regarding the clarity and quality of the presentation. **We would also like to express our sincere thanks to the area chairs for putting in the extra effort to oversee the review process.**

The latest revision of our paper has been uploaded, addressing all comments and queries raised by the reviewers. Edits in the PDF have been highlighted in blue. Below, we provide a summary of the changes made to our work.

------------------------------------------------------------------------------------------------------------------------------------------------------------

## **Writing:**

**Improvements** — Thanks to the suggestions of Reviewer e9oy.

- We have revised the manuscript to **correct grammar** and improve clarity.

## **Theoretical:**

**Scaling Factor** — Thanks to the suggestions of Reviewers N9KR, hVQj, and rfdZ.

- We have added the logarithmic scaling vs. square-root scaling proof in **Proposition B.1**.

**Theoretical Proofs** — Thanks to the suggestions of Reviewers 7Cky and e9oy.

- We have added the $L$-smoothness assumption, the step-size bound, the **PL condition**, and the corrected Lyapunov-based descent argument. These updates are now reflected in **Theorem 3.3, B.4 and B.5**.

- We have added a corrected version of **Lemma A.1** in LogLU proofs.

- In addition, for **Lemma 3.1**, we have corrected the term from **$tG$ to $tG^2$**.

## **Experimental:**

**Hyperparameter Settings** — Thanks to the suggestions of Reviewers 7Cky, rfdZ, and e9oy.

- We have added the **training time and memory usage** results to **Table 5** and included the corresponding discussion in **lines 837–842**.

- We have added the **additional optimizers and activation results** to the activation study, and these updates are now reflected in **Table 3**.

- We have added the additional experimental setup for ImageNet in **lines 230–233** (Section 4.1). The corresponding code **(ViT-S/16)** is provided in the supplementary file.

Best regards,

Authors of submission 20654

---

### Meta-Review · Area_Chair_Zwqd · 2026-01-05

**Summary:**

This paper introduces a logarithmic optimization framework comprising a new activation function, LogLU, and an adaptive optimizer, ZenGrad (and its momentum variant), which utilize logarithmic decay scaling.

The reviewers are polarized, with scores ranging from 2 to 8. While Reviewers N9KR (Score 8) and e9oy (Score 6) recommended acceptance citing empirical gains, I find their assessments overlooked critical flaws in soundness and experimental fairness, which were correctly identified by Reviewers 7Cky and hVQj (Scores 2).

The decision is based on three major concerns:

1. **Theoretical Unsoundness:** Reviewer 7Cky identified that Theorem 3.3 is mathematically flawed. Without establishing $L$-smoothness and a step-size bound to control the second-order remainder (and absent convexity or PL conditions), the claimed Lyapunov decrease and subsequent "convergence towards the global minimum" are unjustified. Crucially, I find that despite the revisions, the authors continue to invalidly treat the element-wise $P_t$ as a scalar in the proof, resulting in a fatal dimension mismatch that fundamentally invalidates the theoretical analysis.


2. **Unfair Experimental Comparison:** Reviewer 7Cky noted that the ViT-S/16 ImageNet jump to 78.8% (Table 1) vs. AdamW/Lion (~70–72%) appears exceptionally large. This performance gap is suspicious given the lack of rigorous experimental details and careful hyperparameter tuning for the baseline optimizers. The rebuttal provided no evidence to validate the fairness of these comparisons.

3. **Limited Novelty:** Reviewer hVQj correctly identified LogLU and ZenGrad as minor variations of Softplus and AdaGrad. While the authors argue for improved smoothness citing empirical gains, I find they failed to isolate the specific contribution of the logarithmic term from other implementation choices or offer valid theoretical justification. Consequently, the novelty remains incremental.

**Reviewer Concerns:**

**Adressed:**

1.	Theoretical
The authors revised the framework by adding Proposition B.1 (log vs. square-root scaling) and introducing standard assumptions like $L$-smoothness and the PL condition. Theorem 3.3 and relevant lemmas were rewritten in an attempt to address prior notation and dimension inconsistencies(7Cky and e9oy).

2.	Experimental
The authors expanded the results to include training costs (Table 5) and additional baseline comparisons (Table 3), alongside clarified hyperparameter settings for reproducibility (7Cky, rfdZ, and e9oy).


**Outstanding:**

Despite the revisions made to the theoretical analysis and experimental results, some critical issues remain unresolved.

1. In the response to Reviewer hVQj, the decay coefficient was erroneously transcribed as $\log_e(P_t) + \epsilon$ instead of $\log_e(P_t + 1) + \epsilon$. Based on this incorrect formula, the provided analysis is fundamentally flawed and completely fails to address the concern raised by Reviewer hVQj.

2. The theoretical analysis in Theorem 3.3 remains fatally flawed due to a dimension mismatch. While Equation (2) explicitly defines $P_t$ as "element-wise", implying the adaptive step size $\eta_t$ is a vector in $\mathbb{R}^d$, the proof treats $\eta_t$ as a scalar. Specifically, the derivation invalidly simplifies the descent term to $-\eta_t \|\nabla \mathcal{L}(w_t)\|^2$.

**Reviewer Scores:**

**Reviewer N9KR ( Score : 8  -> Est. 4)**:

This reviewer admitted "Math/other details were not carefully checked," causing them to overlook critical theoretical flaws regarding L-smoothness, step-size bounds, etc. This invalidates the theoretical "strength" underpinning their high score, necessitating a significant downgrade.

**Reviewer e9oy (Score : 6 -> Est. 4)**:

This reviewer explicitly flagged the ViT-S/16 results as "too impressive" and potentially suspicious (W3). The rebuttal failed to dispel this doubt, as the authors did not provide evidence of rigorous hyperparameter tuning for the baselines to justify such a massive performance gap.

**Reviewer 7Cky (Score : 2 -> Est. unchanged)**:

 This reviewer correctly identified fundamental theoretical flaws. The rebuttal failed to rectify the invalid scalar proof for the element-wise algorithm and offered no evidence to resolve valid concerns regarding unfair experimental comparisons.

**Reviewer	hVQj (Score : 2 -> Est. unchanged)**:

 This reviewer correctly flagged the incremental novelty. The rebuttal failed to offer valid theoretical justification, relying on a trivial inequality, and introduced a new transcription error in the decay formula ($\log_e P_t$ instead of $\log_e(P_t+1)$), reinforcing the lack of rigor and failing to resolve the core concern regarding the method's value.

**Reviewer rfdZ (Score : 4 -> Est. unchanged)**:

 This reviewer praised the "rigorous theoretical analysis" but admitted "Math/other details were not carefully checked," leading them to overlook critical flaws in the mathematical derivation. While the authors addressed the specific concern regarding computational overhead by adding Table 5, the fundamental theoretical invalidity justifies maintaining the rejection rating.

---

### Decision · Program_Chairs · 2026-01-26

Reject